# *In vivo* neural activity of electrosensory pyramidal cells: Biophysical characterization and phenomenological modeling

**Amin Akhshi**, **Michael G. Metzen**, **Maurice J. Chacron**, **Anmar Khadra**\*

Department of Physiology, McGill University, Montreal, Quebec, Canada

\* anmar.khadra@mcgill.ca

## Abstract

Burst firing is an important property of neuronal activity, thought to enhance sensory encoding. While previous studies show significant differences in burst firing between *in vivo* and *in vitro* conditions, how burst firing contributes to neural coding *in vivo* and how it is modulated by underlying biophysical mechanisms when neurons are under active synaptic bombardments remains poorly understood. Here, we combined intracellular recordings and computational modeling to investigate how cellular and synaptic mechanisms can explain the *in vivo* firing activity of electrosensory lateral line lobe (ELL) pyramidal cells in *Apteronotus leptorhynchus*. We developed a biophysically detailed compartmental model incorporating voltage-gated currents, NMDA receptor-mediated calcium ($Ca^{2+}$) influx, $Ca^{2+}$-activated SK channels, $Ca^{2+}$ mobilization, and stochastic synaptic inputs to reproduce *in vivo* firing activities of ELL pyramidal cells. Specifically, using bifurcation analysis, we identified dynamical transitions between quiescent, tonic, and bursting regimes, governed by interactions among SK conductance, NMDA receptor activation, and applied current. Model parameters were optimized against *in vivo* data, accurately reproducing action potential waveforms and temporal dynamics, including characteristic bimodal interspike interval distributions reflecting intra- and inter-burst intervals. We further developed a modified Hindmarsh-Rose model incorporating dual adaptation variables and stochastic noise. This simplified phenomenological model successfully captured burst firing comparable to that observed in the biophysical model and recorded data, while replicating diverse firing patterns observed across the population. Finally, parameter sensitivity analysis revealed slow adaptation dynamics and noise intensity as key determinants of spiking variability within cells. Overall, our modeling results demonstrate that *in vivo* bursting arises from synergistic interactions between intrinsic conductances (e.g., NMDA-SK coupling), $Ca^{2+}$ mobilization, and synaptic stochasticity, offering a potential reconciliation for discrepancies with *in vitro* firing activity. The

**Data availability statement:** Code used to run simulations, analyze data, and generate manuscript figures is available on GitHub (https://github.com/aminakhshi/spc_mobjective).

**Funding:** This research was funded by the Fonds de Recherche du Québec – Nature et Technologies (FRQNT) team grant to MJC and AK, a Canadian Institutes of Health Research (CIHR) grant to MJC, and Natural Sciences and Engineering Research Council of Canada (NSERC) discovery grants to MJC and AK. AA was supported by the NSERC-CREATE in Complex Dynamics, Healthy Brains Healthy Lives (HBHL) and the FRQNT fellowships. The funders had no role in study design, data collection and analysis, decision to publish, or preparation of the manuscript.

**Competing interests:** The authors have declared that no competing interests exist.

models provide mechanistic insights into how background synaptic activity modulates burst firing.

---

## Author summary

In the brain, neurons often fire bursts, i.e., brief clusters of action potentials separated by quiescent periods; these bursts are thought to help signal important information. However, most of what we know about how bursts are generated comes from experiments on brain slices under laboratory conditions, where neurons are isolated from the continuous background synaptic activity present in the intact brain. In our study, we sought to understand how neurons behave in their natural, active environment. We focused on a group of sensory neurons in a type of weakly electric fish, which offers a well-characterized model to study how sensory modalities process incoming information that results in perception and behavior. By combining intracellular recordings from living animals with a computational modeling approach, we showed that burst firing *in vivo* is shaped by both internal cellular mechanisms and the ongoing synaptic input the cell receives. We also developed a simplified phenomenological version of our model that, due to its simpler structure, can be used in future studies to potentially explore network-level dynamics. Our work sheds light on the underlying mechanisms that differentiate firing patterns observed *in vitro* versus *in vivo*, and provides computational tools for better understanding how the brain processes information in real-world conditions.

## Introduction

Understanding how neurons in sensory systems process incoming information to drive perception and behavior remains a fundamental challenge in neuroscience. In particular, this understanding is complicated by the fact that even neurons of the same anatomical or functional type can still display diverse and complex spiking activity patterns in response to common stimuli [1–3], which are thought to enhance the encoding of stimulus-relevant information [4–6]. Growing evidence, however, suggests that such firing patterns can differ significantly between *in vivo* and *in vitro* conditions [7–12]. Indeed, in the intact behaving brain, neurons are subject to ongoing background synaptic activity that drives them to a high-conductance state, characterized by enhanced membrane potential fluctuations and altered input–output relationships [9,13–16]. This state enhances variability and synaptic input responsiveness [9,14,15,17]. Such fluctuations can improve temporal precision and facilitate efficient sensory information coding [5,18–21], and are therefore thought to contribute to changes in the timescale of neural responses and synchronization characteristics [9,22–24]. Together, these effects underscore the critical importance of understanding how *in vivo* conditions influence neuronal dynamics, a topic that remains to be systematically explored.

One of the most prominent examples of a neural system exhibiting *in vivo* vs. *in vitro* discrepancies is found in pyramidal cells within the electrosensory lateral line lobe (ELL) of weakly electric fish [12,25–27]. The electrosensory system of these fish offers a well-characterized neural circuit, making it advantageous for studying how *in vivo* conditions affect neural activity patterns [28–31]. These animals generate a continuous electric organ discharge (EOD) used for navigation and communication, and detect perturbations of this signal via electroreceptor afferents (EAs) distributed across the skin, which relay sensory input to pyramidal cells within the ELL [32] that project to the midbrain [23,31,33–36]. A defining feature of ELL pyramidal cells is their ability to fire action potentials in the form of bursts (i.e., clusters of action potentials separated by quiescent periods). *In vitro* studies of ELL pyramidal cells have revealed that burst firing relies on a somato-dendritic interplay in which the backpropagating action potentials from the soma to the apical dendrites generate bursts typically containing 5–8 spikes [37–39]. This process is associated with a progressive increase in the amplitude of depolarization afterpotentials (DAPs) at the soma and shortening interspike intervals (ISIs) until the burst terminates due to failure of dendritic spike backpropagation, followed by a large burst afterhyperpolarization (AHP) [40–42]. Computational modeling studies of ELL pyramidal cells have identified this burst mechanism as ghostbursting [39,41]. Unlike classical bursting models, where bursts arise from slow alternations between quiescent and oscillatory states, ghostbursting emerges from the interaction between somatic and dendritic spikes [39]. The underlying dynamics are governed by periodic orbit bifurcations, with burst termination occurring when trajectories are reinjected near the ghost of a saddle-node bifurcation of fixed points in phase space. This reinjection leads to long inter-burst intervals and irregular, short bursts, often comprised of a few spikes, sometimes as few as two in the form of doublets [39,41,43,44]. The *in vivo* recordings, on the other hand, show markedly different firing patterns, typically consisting of only 2-3 spikes per burst on average, with no evidence of a somatic DAP or burst AHP [25,27,43,45]. Notably, pharmacological blockade of intracellular $Ca^{2+}$ using BAPTA restores *in vitro*-like burst firing patterns *in vivo* [25,46–48], implicating the important role of $Ca^{2+}$ dynamics and $Ca^{2+}$-activated SK channels as critical modulators of burst dynamics under *in vivo* conditions. In parallel, reducing ongoing background synaptic drive provided by descending feedback decreases ELL pyramidal-cell firing [43,49], indicating that $Ca^{2+}$-dependent intrinsic mechanisms and feedback-mediated synaptic input jointly shape the *in vivo* burst regime. The disparity between the *in vivo* and *in vitro* recordings suggests that detailed computational models are required to account not only for additional intrinsic ionic conductances expressed *in vivo* but also for extrinsic factors that influence spiking activity.

In this study, we combined intracellular *in vivo* recordings of ELL pyramidal cells in *Apteronotus leptorhynchus* with computational modeling to investigate how biophysical mechanisms such as $Ca^{2+}$ dynamics, SK and NMDA receptor currents, as well as stochastic synaptic inputs, can explain discrepancies between neuronal firing activities seen *in vitro* and *in vivo*. First, we built a detailed biophysical model and constrained its parameter sets to accurately reproduce both the waveform and the temporal dynamics of action potentials. We subsequently employed bifurcation analysis to systematically examine the influence of key parameters on the critical transitions between quiescent, tonic, and burst firing regimes. Next, we developed a phenomenological model based on the Hindmarsh-Rose (HR) model and demonstrated that the modified HR model accurately reproduces the spiking activity of ELL pyramidal cells, providing a computationally efficient framework for exploring the spiking activity of these cells and making it suitable for future applications to population-level coding in neural circuits.

## Results

The goal of this study was to use computational modeling to investigate how cellular and synaptic mechanisms can explain experimentally observed differences in firing activity of ELL pyramidal cells *in vitro* and *in vivo*. ELL pyramidal cells receive direct ascending inputs from electroreceptor afferents (EAs; Fig 1A, black arrow) as well as substantial descending feedback (Fig 1A, orange arrows) [33,34]. These inputs regulate firing activity through a combination of synaptic

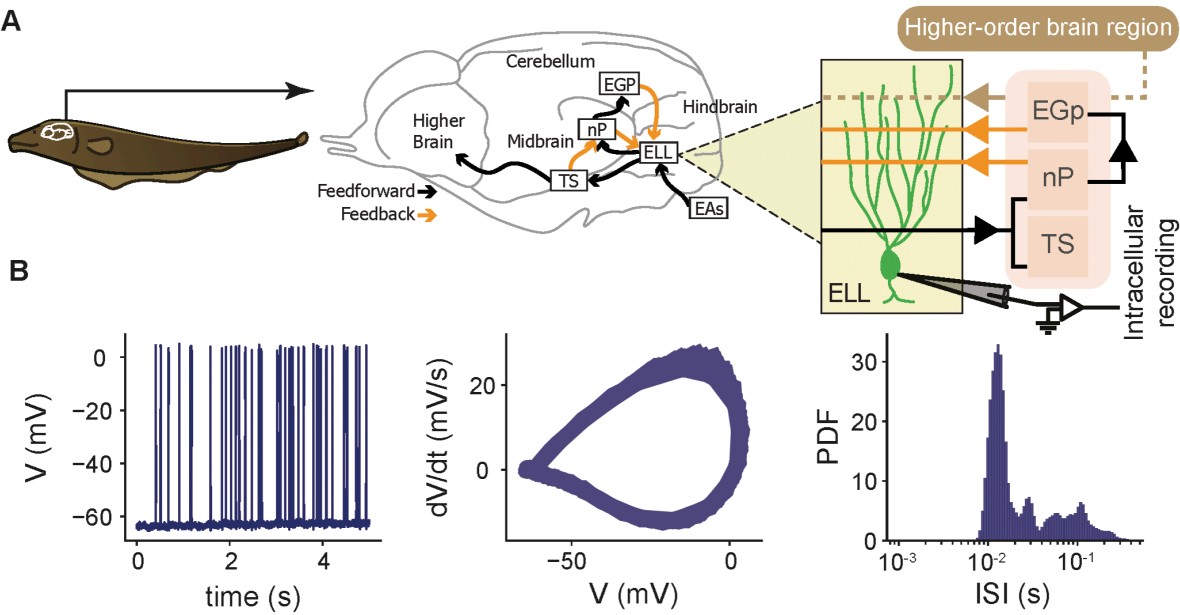

**Fig 1. Circuit and single cell electrical properties of electrosensory lateral line lobe (ELL) pyramidal cells. (A)** Schematic of the neural circuitry underlying the electrosensory processing in the weakly electric fish (left). Brain circuitry (middle) showing how the ELL receives feedforward input from electrosensory afferents (EAs) and feedback signals from higher brain areas, including the eminentia granularis posterior (EGp), nucleus praeminentialis (nP), and torus semicircularis (TS). Intracellular recordings were obtained from pyramidal neurons within the ELL to investigate their electrical activity (right). **(B)** Intracellular membrane potential trace recording for 5 sec (left), phase plane plot of action potential cycles in the voltage trace (middle), and interspike interval (ISI) distribution (right) for a representative recording.

mechanisms involving both glutamatergic excitation (e.g., mediated in part by NMDA receptors) [23,35,50] and feedback-activated inhibition [51,52]. We performed single-unit intracellular recordings *in vivo* from $n = 32$ pyramidal cells in the ELL across $N = 8$ fish under baseline conditions, i.e., in the absence of external stimulation (Fig 1A, right; see Methods for details). In this condition, the animal's own electric organ discharge (EOD) is present but unmodulated, which does not provide sensory input by itself unless perturbed by objects or conspecifics [30,47,53]. As detailed in the Methods, immobilized animals were housed individually in isolated tanks to prevent such perturbations. Fig 1B shows the membrane potential trace for a representative neuron (Fig 1B, left; 5 sec duration) displaying spiking activity characterized by periods of rapid, clustered spiking, consistent with burst firing. Notably, and in contrast to firing patterns observed *in vitro* [37–39,41], these *in vivo* bursts consisted of fewer spikes and lacked prominent DAPs and distinct burst-specific AHPs. Consistent with the clustered spiking observed in the voltage trace (Fig 1B, left), the interspike interval (ISI) distribution for this neuron was bimodal (Hartigan's dip test, *p*-value = 0.00402) (Fig 1B, right), with a prominent peak at short ISIs (<~ 12 ms) corresponding to the intra-burst interval and a peak at larger values (~ 100 ms) corresponding to the inter-burst interval. To quantify burst firing across the population, we defined bursts based on a threshold set at the local minimum between these two ISI modes to separate spike trains into burst and isolated spikes for each cell [54]; a burst was identified as any sequence of two or more spikes in which all consecutive ISIs fell below this threshold (S1A Fig; see Methods). Unlike approaches that treat bursts as unitary events, we explicitly accounted for burst structure, including the number of spikes within each burst, and computed the fraction of spikes identified as belonging to bursts.

Consistent with previous studies [21,43,47,55], we observed that the mean firing rate across the recorded population ranged between 5–60 Hz in the absence of stimulation (S1B Fig, left), and that spike train statistics including the tendency for burst firing varied considerably between cells (S1B Fig, right).

**A biophysical model of ELL pyramidal cells accurately reproduces the *in vivo* action potential characteristics**

To gain understanding as to how different mechanisms can account for differences between *in vivo* vs *in vitro* bursting dynamics in ELL pyramidal cells, we developed a new conductance-based biophysical model, partially inspired by previous studies [25,39], that incorporates the flux balance model of $Ca^{2+}$ mobilization (Fig 2A; see Methods). As done previously [39], the intrinsic bursting patterns of ELL pyramidal cells were captured by representing the soma and dendrites as two coupled isopotential compartments linked by a resistive current scaled by the soma-to-dendrite area ratio ($\kappa$), with each compartment containing distinct ionic conductances (Fig 2A; see Methods). The two compartments included fast $Na^+$ ($I_{Na,S}, I_{Na,D}$, respectively) and delayed rectifier $K^+$ ($I_{Dr,S}, I_{Dr,D}$, respectively) currents, both responsible for generating action potentials. The dendritic compartment, however, included additionally $Ca^{2+}$-activated small conductance $K^+$ current ($I_{SK}$) that hyperpolarizes the membrane voltage in response to intracellular $Ca^{2+}$ increases and helps modulate burst termination, as well as a $Ca^{2+}$ current due to NMDA receptors ($I_{NMDA}$), the primary source of $Ca^{2+}$ influx mediating dendritic depolarization (Fig 2A; dendrite). The kinetics of the NMDA receptors was based on a Markov chain model [56] consisting of three closed states ($C_0, C_1, C_2$), one conducting open state ($O$) and one desensitized state ($D$) (see Methods), allowing the receptors to generate currents that are voltage-dependent due to $Mg^{2+}$ block ($[Mg^{2+}]_o$). During depolarization, the magnesium block is relieved, allowing $Ca^{2+}$ influx into the cell. This creates a regenerative depolarizing current that sustains dendritic spikes and amplifies burst generation. The flux-balance model for $Ca^{2+}$ mobilization in the dendritic compartment was based on the Li-Rinzel formalism, accounting for $Ca^{2+}$ entry through NMDA receptors, $Ca^{2+}$ buffering in the cytosol to keep it at low concentration, IP3 receptor flux via $Ca^{2+}$-induced $Ca^{2+}$-release mechanism from the endoplasmic reticulum (ER), SERCA flux that pumps $Ca^{2+}$ back into the ER for sequestration, PMCA flux that pumps $Ca^{2+}$ out of the cell and leak fluxes through plasma and ER membranes [57,58]. The resulting model generates cytosolic $Ca^{2+}$ transients that activate $I_{SK}$, contributing to the termination of bursts by inducing afterhyperpolarization (Fig 2A; right). The ER $Ca^{2+}$ stores, on the other hand, can deplete and recover, introducing temporal variability in burst initiation and duration. To account for the contribution of synaptic inputs to burst generation, we included a synaptic input current ($I_{syn}$) in the dendritic compartment to represent the stochastic background excitatory and inhibitory pre-synaptic activity [24,59,60]. Under stochastic synaptic input, the model generated voltage dynamics and $Ca^{2+}$ transients that are very similar to those observed *in vivo* (S2 Fig).

To assess the performance of our model and constrain its parameters to physiologically relevant ranges *in vivo*, we fitted action potential features from simulations to those extracted from intracellular recordings (see Methods), selecting parameter values that reproduced experimental features and preserved model realism. Feature extraction followed the Allen Institute's protocols for electrophysiological data processing, which include features such as voltage threshold for spiking, peak amplitudes, trough amplitudes, voltage at midpoint of the upstroke phase, voltage at midpoint of the downstroke phase, and amplitude of afterdepolarization potential (Fig 2B; highlighted in pink). To isolate individual action potentials, we extracted segments of the voltage time series centered around the peak amplitudes within a 8 ms window (i.e., 4 ms before and after the peak; see Methods). A comparison of extracted features from experimental recordings (dark blue) and model simulations (light blue) revealed a close match, as evidenced by the overlaid simulated spikes on recorded spikes for a representative ELL pyramidal cell (Fig 2C). The model accurately reproduces the shape and defining characteristics of action potentials. This agreement was further quantified by statistical comparisons of each feature between the experimental data and model simulations using two-sample t-tests, Mann–Whitney U tests, and Kolmogorov–Smirnov (KS) tests. All three tests confirmed no significant differences for any of the examined features (Fig 2D; threshold$v$: $p_t = 0.054$, $p_{MW} = 0.48$, $p_{KS} = 0.34$; upstroke$v$: $p_t = 0.152$, $p_{MW} = 0.48$, $p_{KS} = 0.22$; peak$v$: $p_t = 0.694$, $p_{MW} = 0.82$, $p_{KS} = 0.99$; downstroke$v$: $p_t = 0.395$, $p_{MW} = 0.54$, $p_{KS} = 0.68$; trough$v$: $p_t = 0.502$, $p_{MW} = 0.54$, $p_{KS} = 0.32$; adp$v$: $p_t = 0.283$, $p_{MW} = 0.54$, $p_{KS} = 0.36$). Furthermore, to assess the model's performance across the population, we compared the mean value of each feature between the model and experimental data across all recorded cells (S3 Fig; linear fit: $r_{threshold} = 0.98$, $p_{threshold} = 3.7 \times 10^{-23}$; $r_{upstroke} = 0.97$, $p_{upstroke} = 4.6 \times 10^{-21}$; $r_{peak} = 0.99$, $p_{peak} = 1.0 \times 10^{-27}$;

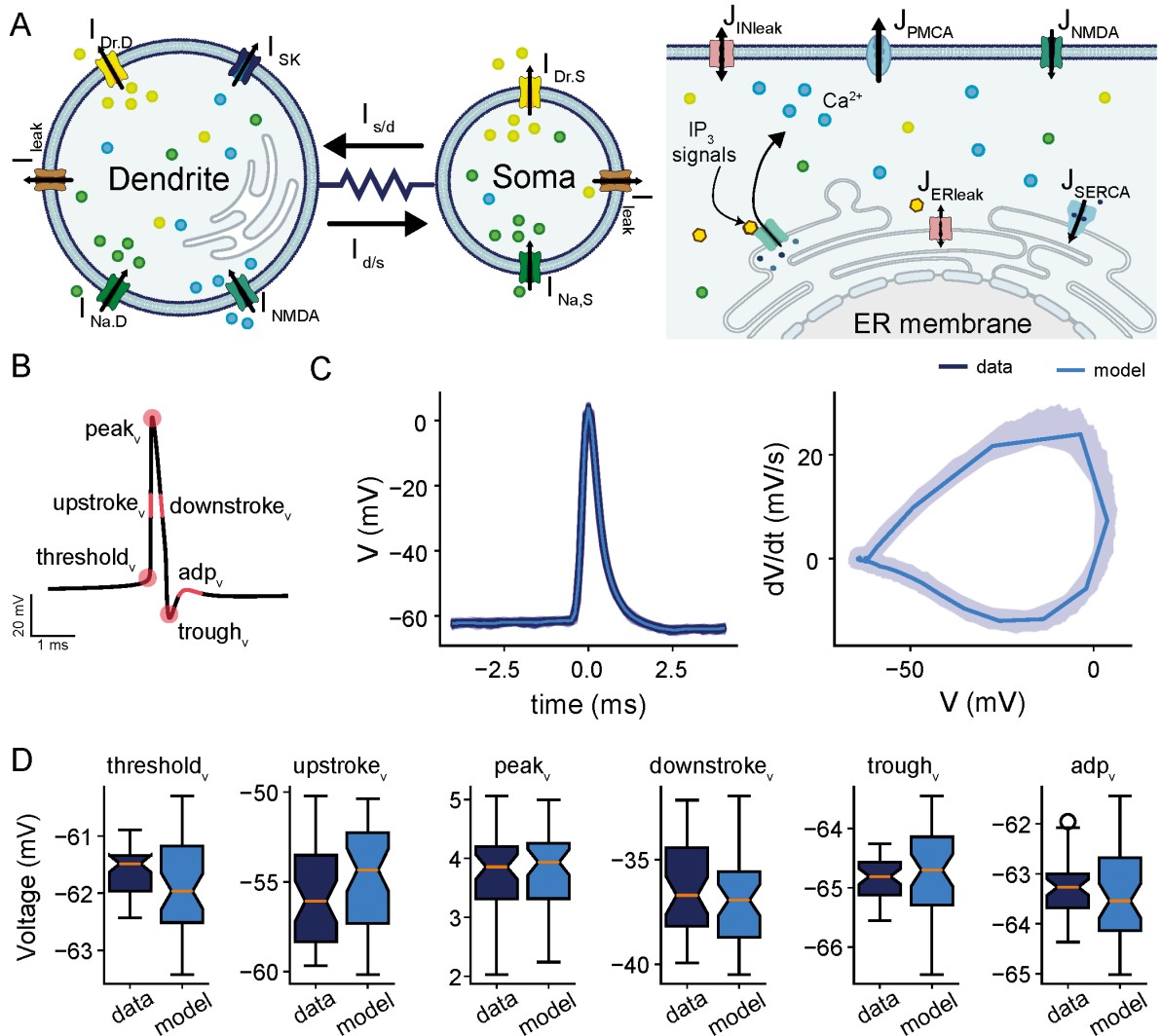

**Fig 2. Comparison of electrophysiological features between intracellular recording of a given ELL pyramidal cell and a model simulation.**
**(A)** Schematic of the model showcasing the two-compartments of the cell: soma and dendrite (left), and highlighting the various ionic currents expressed in each compartment, including fast Na$^+$ ($I_{Na,i}$), delayed rectifier K$^+$ ($I_{K,i}$), small-conductance K$^+$ ($I_{SK}$), NMDA ($I_{NMDA}$) and leak ($I_{leak}$) currents in the soma ($i = S$) and dendrite ($i = D$). The kinetics of NMDA receptors are governed by a Markov model (see Methods). The schematic of Ca$^{2+}$ dynamics within the dendritic compartment following the flux-balance model is also shown (right). It displays all fluxes involved in regulating Ca$^{2+}$ mobilization across plasma and ER membranes in the dendritic compartment. These fluxes include Ca$^{2+}$ entry through NMDA receptors ($J_{NMDAR}$) and IP3 receptors ($J_{IP3R}$), Ca$^{2+}$ efflux through SERCA ($J_{SERCA}$) and PMCA ($J_{PMCA}$) pumps, and leak across both membranes. **(B)** Schematic of an action potential with the extracted electrophysiological features (highlighted in pink) for model fitting; that includes from left to right: spike threshold (threshold), midpoint of the upstroke phase (upstroke$_v$), peak amplitudes (peak$_v$), midpoint of the downstroke phase (downstroke$_v$), trough amplitudes (trough$_v$) and amplitude of afterdepolarization potential (adp$_v$). **(C)** Extracted action potentials (left) and action potential cycles (right) obtained from one recorded trace (5 sec; shaded dark blue) and the average action potential and average action potential cycle of all spikes obtained from a single model simulation overlaid on top (5 sec; light blue). **(D)** Box plots of electrophysiological features obtained from all action potentials extracted from the experimental recording and model simulation in C. Panels from left to right: threshold$_v$, upstroke$_v$, peak$_v$, downstroke$_v$, trough$_v$ and adp$_v$. Statistical comparisons were performed using two-sample t-tests, Mann–Whitney U tests, and Kolmogorov–Smirnov tests. No significant differences were found for any feature (threshold$v$: $p_t = 0.054$, $p_{MW} = 0.48$, $p_{KS} = 0.34$; upstroke$v$: $p_t = 0.152$, $p_{MW} = 0.48$, $p_{KS} = 0.22$; peak$v$: $p_t = 0.694$, $p_{MW} = 0.82$, $p_{KS} = 0.99$; downstroke$v$: $p_t = 0.395$, $p_{MW} = 0.54$, $p_{KS} = 0.68$; trough$v$: $p_t = 0.502$, $p_{MW} = 0.54$, $p_{KS} = 0.32$; adp$v$: $p_t = 0.283$, $p_{MW} = 0.54$, $p_{KS} = 0.36$). For adp$_v$ estimation, spikes without a detected ADP were omitted.

$r_{\text{downstroke}} = 0.98$, $p_{\text{downstroke}} = 1.0 \times 10^{-26}$; $r_{\text{trough}} = 0.98$, $p_{\text{trough}} = 2.7 \times 10^{-24}$; $r_{\text{adp}} = 0.97$, $p_{\text{adp}} = 1.7 \times 10^{-26}$). The results showed points clustering tightly around the identity line, indicating strong agreement and demonstrating that the model accurately captures the average feature characteristics across the cell population.

Overall, these results show that the biophysical two-compartment model effectively captures the physiological characteristics of *in vivo* action potentials in ELL pyramidal cells.

### The model captures burst dynamics and spiking variabilities observed across the ELL pyramidal cell population *in vivo*

Next, we assessed the ability of the model to reproduce the temporal firing dynamics of ELL pyramidal cells *in vivo* by comparing key spiking activity features between the data and model simulations obtained after parameter optimization (see Methods). Fig 3A compares representative voltage traces (top panels) and corresponding ISI distributions (bottom

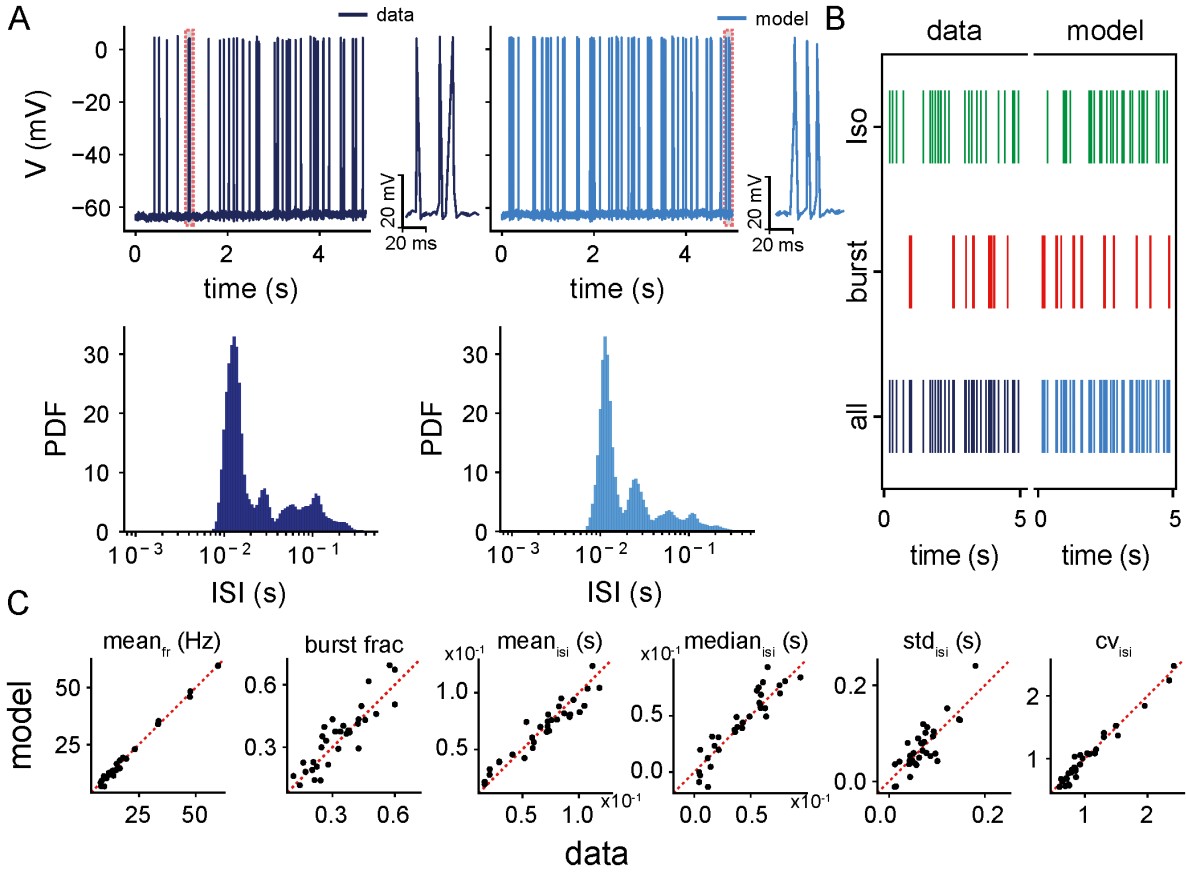

**Fig 3. Comparison of spike train features between intracellular recordings of a population of ELL pyramidal cells and their fitted model simulations.** **(A)** Membrane potential traces (top) and interspike interval (ISI) distribution (bottom) of a representative ELL pyramidal cell (dark blue, left) and its fitted model simulations (light blue, right). The recorded and simulated traces are 5 seconds long, showing sequences of bursts interspersed with isolated spikes and quiescent periods. Insets show zoomed-in examples of individual bursts from the recorded and simulated traces, highlighting the close match in spike shape and temporal organization within bursts. **(B)** Raster spike train plots from same recording in A (left) and their corresponding fitted simulations (right), separated into isolated spikes (green), burst spikes (red), and all spikes combined (blue). The model replicates the proportions and temporal organization of burst and isolated spikes observed in the recordings. **(C)** Scatter plots comparing key spike train features between recordings and simulations for all ELL pyramidal cells ($n = 32$). Panels from left to right: mean firing rate (mean$_{fr}$; $r = 0.99$, $p = 4.3 \times 10^{-35}$), burst fraction (burst frac; $r = 0.88$, $p = 2.7 \times 10^{-11}$), mean (mean$_{isi}$; $r = 0.95$, $p = 1.1 \times 10^{-16}$), median (median$_{isi}$; $r = 0.92$, $p = 3.3 \times 10^{-14}$), standard deviation (std$_{isi}$; $r = 0.83$, $p = 2.6 \times 10^{-09}$) and the coefficient of variation (CV$_{isi}$; $r = 0.98$, $p = 1.3 \times 10^{-23}$) of ISIs, respectively.

panels) between the example recording data (left) and the model simulation fitted to that data (right). Both traces exhibit qualitatively similar spiking patterns, including sequences of bursts interspersed with isolated spiking events. To better illustrate burst structure, the insets in the top panels show zoomed-in views of individual bursts from the recorded and simulated traces, highlighting the close match in the shape and temporal organization of spikes within bursts. The corresponding ISI distributions show a close alignment (Kolmogorov-Smirnov test: $D = 0.335$, $p = 0.635$), accurately preserving the characteristic bimodality observed in the data. In both distributions, the left mode primarily corresponds to fast spiking events within bursts (intra-burst spikes), while the second mode at longer intervals reflects the timing between bursts or isolated spikes. Furthermore, raster plots comparing data and model outputs demonstrate that the model accurately reproduces the qualitative structure of these bursts and isolated spikes (Fig 3B). Interestingly, the most commonly observed burst event patterns in both the data and the model included a few spikes within each burst, consistent with previous studies [43,45]. The close match in both ISI distributions and spike timing structure highlights the model's ability to capture key temporal dynamics of both burst firing and regular spiking activity *in vivo*.

To quantify the overall performance of the model in reproducing the temporal firing activities of all recorded ELL pyramidal cells ($n = 32$), we systematically compared key spike train features generated by these cells to those generated by model simulations; that included: the mean firing rate, burst fraction, mean and median of ISI, the standard deviation of ISI, and the coefficient of variation for each recorded ELL pyramidal cell compared with the representative model neuron obtained by the best fit for that cell (Fig 3C). Burst fractions for model simulations were obtained using the same method as for the data (see Methods). The pairwise comparison of all features showed a strong linear correspondence between experimental and simulated values across the population. All points clustered closely around the identity line, indicating that the model accurately reproduced cell-specific variability in firing activity (Fig 3C; linear fit: mean firing rate ($r = 0.99$, $p = 4.3 \times 10^{-35}$), burst fraction ($r = 0.88$, $p = 2.7 \times 10^{-11}$), mean ($r = 0.95$, $p = 1.1 \times 10^{-16}$), median ($r = 0.92$, $p = 3.3 \times 10^{-14}$), standard deviation ($r = 0.83$, $p = 2.6 \times 10^{-09}$), and coefficient of variation ($r = 0.98$, $p = 1.3 \times 10^{-23}$) of ISI distributions, respectively). This further indicates that the temporal structure of spiking activities *in vivo* across the recorded population is well-preserved by the model.

Taken together, these results highlight that the biophysical model accurately reproduces variability in the burst firing dynamics of ELL pyramidal cells *in vivo*.

## Bifurcation analysis of biophysical model elucidates how intrinsic mechanisms affect spiking activity

We next investigated the deterministic dynamics of the model by exploring the effects of varying key model parameters including the applied current ($I_{app}$) and the maximal conductances of $I_{SK}$ and $I_{NMDA}$ ($g_{SK}$ and $g_{NMDA}$, respectively) to understand how they affect spiking activity. Our one-parameter bifurcation analysis revealed that varying $I_{app}$ produces four distinct firing regimes: two quiescent (i.e., no action potential firing), one tonic firing (i.e., periodic firing of action potentials) and one ghostbursting regime; these regimes are separated by different types of bifurcation points (Fig 4A). For low values of $I_{app}$, the model exhibits three branches of equilibria: a stable one (bottom branch; solid orange line) corresponding to the quiescent state of the cell, and two unstable ones (middle and top branches; dashed black lines). Together, the three branches form an s-shaped manifold with two saddle nodes: one on the right (labeled SN1) where the bottom two branches merge, forming the right boundary of the left quiescent regime, and one on the left (labeled SN2) where the top two branches merge. The presence of three coexisting steady states on the left of SN1 enables the model to generate (among other things; see below) single spikes in response to prolonged suprathreshold constant current injections (S4A Fig). These spikes arise from type IV excitability [61,62], triggered when trajectories cross the threshold defined by the stable manifold of the saddle fixed points (the middle equilibria).

At larger $I_{app}$, the upper unstable branch stabilizes through a supercritical Hopf bifurcation (HB; Fig 4A), from which a branch of stable periodic orbits emerges (green solid lines). HB divides the parameter space into a tonic firing regime on the left and a second quiescent regime (known as the depolarization block) on the right. In the tonic firing regime, the

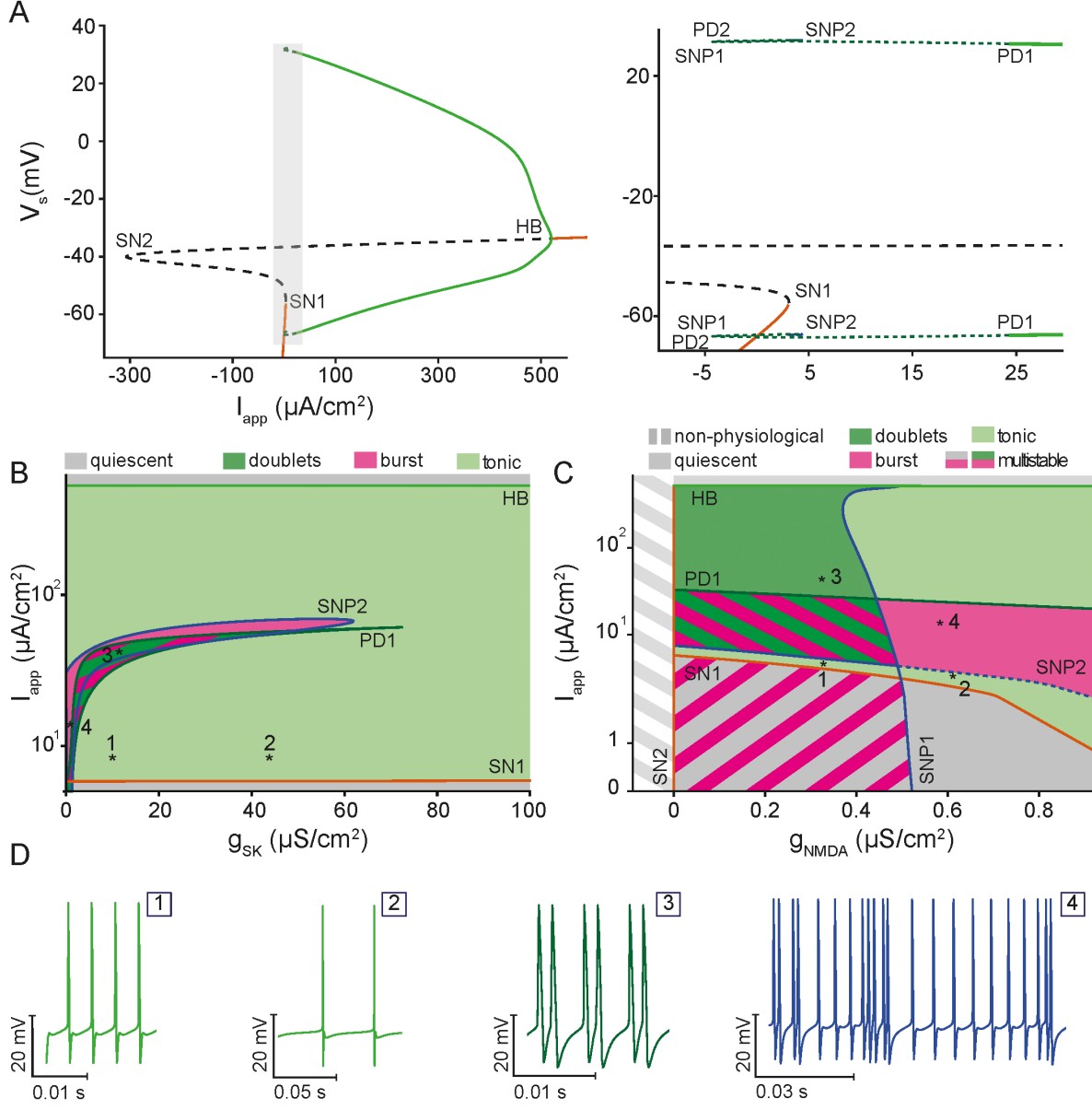

**Fig 4**. **Bifurcation analysis of the deterministic biophysical model reveals the full dynamical behavior of ELL pyramidal cells. (A)** One-parameter bifurcation diagram of the voltage variable ($V$) of the biophysical model with respect to the applied current ($I_{app}$), illustrating the various regimes of behavior corresponding to branches of stable equilibria (orange line), branches of unstable equilibria (dashed black line), a branch of stable periodic orbits (green lines) and a branch of stable bursting orbits (dashed green line). Within the range of $I_{app}$ considered, the model undergoes several types of bifurcation, including 2 saddle-node bifurcations (SN1, SN2), 1 Hopf bifurcation (HB), 2 saddle-node bifurcations of periodic orbits (SNP1, SNP2) and 2 period-doubling bifurcations (PD1, PD2), some of which define the boundaries of the various regimes of behavior identified. The inset provides magnified views of the area enclosed by the highlighted region in the main figure. **(B)** The two-parameter bifurcation diagram of the biophysical model with respect to $I_{app}$ and the maximum conductance of SK channels ($g_{SK}$), showing the distinct dynamical regimes of the model, including quiescence, tonic firing and doublet/chaotic ghostbursting. The region bounded by PD1 is multistable, featuring coexisting distinct attractors. **(C)** The two-parameter bifurcation diagram of the biophysical model with respect to $I_{app}$ and the maximum conductance of NMDA receptors ($g_{NMDA}$), showing the distinct dynamical regimes of the model, including quiescence, tonic firing and doublet/chaotic ghostbursting. **(D)** A sample of three principal firing patterns: tonic spiking at high (1), and low frequencies (2), ghostbursting in the form of doublets (3) and chaotic ghostbursting (4) at different values of $I_{app}$. These patterns demonstrate the rich repertoire of activity captured by the biophysical model.

firing frequency of limit cycles increases with $I_{app}$. At lower values of $I_{app}$ near SN1 (Fig 4A, inset), the periodic orbit undergoes a period-doubling (PD1) bifurcation, leading to the formation of the ghostbursting regime on the left of PD1. Further decreasing $I_{app}$ gives rise initially to a secondary period-doubling (PD2) bifurcation and two saddle-node of periodics bifurcations (labeled SNP1 and SNP2) followed by a cascade of infinite number of bifurcations that become increasingly hard to discern (Fig 4A, inset); this cascade reflects the onset of doublet firing and chaotic dynamics and the emergence of multistable bursting orbits. Interestingly, on the left of SN1, the model shows bistability, where the quiescent state coexists with bursting orbits, which themselves represent a multistable regime, further highlighting the complex nature of this regime (S4B Fig).

To explore how these distinct dynamic regimes are modulated by intrinsic mechanisms, we next conducted a two-parameter bifurcation analysis in which $I_{app}$ (vertical axis) was varied in conjunction with the maximal conductances of either $I_{SK}$ ($g_{SK}$), or $I_{NMDA}$ ($g_{NMDA}$) (Fig 4B and C, respectively). This was done by tracking the bifurcation points identified in the one-parameter bifurcation with respect to $I_{app}$ (Fig 4A) as a second parameter ($g_{SK}$, $g_{NMDA}$) was varied. In the case of the $g_{SK}$, the continuation in the two parameter space identified four distinct regions of behavior, including no spiking (Fig 4B; gray, quiescent bottom and depolarization block top), doublet firing (Fig 4B and D:3; dark green), tonic firing (Fig 4B and D:1-2; light green), and bursting (Fig 4B and D:4; purple). Specifically, our results showed that increasing $g_{SK}$ gradually shifts the boundaries of the bursting region towards higher $I_{app}$ values, indicating that stronger $g_{SK}$ modulates the spiking activity in the model by reducing the firing rate. Indeed, at intermediate $I_{app}$ values, increasing $g_{SK}$ can shift the cell from bursting to tonic firing with a lower spiking frequency (Fig 4D:1-2). On the other hand, larger values of $I_{app}$ give rise to transition from tonic firing to either doublet firing or bursting, depending on the value of $g_{SK}$, followed by a return to tonic firing with high frequencies until crossing HB, beyond which the cell ceases to fire at the depolarization block region.

Similarly, the dynamical transitions obtained by continuing the bifurcation points from Fig 4A in a two-parameter space defined by $I_{app}$ and $g_{NMDA}$ revealed very rich dynamics (Fig 4C). As was the case for $g_{SK}$, several distinct regions of behavior were identified, ranging from quiescence (gray, representing no spiking regions as well as a non-physiological region with unrealistic conductance values), doublet firing (dark green), bursting (purple), and tonic spiking (light green). The bifurcation boundaries, corresponding to SN1, SN2, HB, PD1, SNP1 and SNP2, display a more complex structure compared to the continuation for $g_{SK}$, reflecting intricate interactions between $I_{app}$ and $g_{NMDA}$. Notably, increasing $g_{NMDA}$ significantly alters the $I_{app}$ thresholds at which transitions between firing regimes occur. For example, the $I_{app}$ range that allows for bursting (purple region) is highly sensitive to $g_{NMDA}$, expanding and shifting as $g_{NMDA}$ increases. This sensitivity may enable synaptic inputs to more effectively (and robustly) induce bursting. Indeed if $I_{SK}$ and $I_{NMDA}$ are blocked ($g_{SK} = 0$, $g_{NMDA} = 0$), the model exhibits prolonged bursts with prominent depolarizing afterpotentials (DAPs), reverting to dynamics similar to the original *in vitro* ghostbursting model (S4C Fig).

Taken together, these results showcase how adjusting $g_{SK}$ or $g_{NMDA}$ in conjunction with the applied current governs the transitions between quiescence, tonic firing, and bursting. The $Ca^{2+}$-dependent transitions to bursting seen here are consistent with previous work showing that irregular bursting can arise when slow intracellular $Ca^{2+}$ dynamics interact with the fast spiking subsystem [63]. In our model, dendritic $Ca^{2+}$ influx through NMDA receptors and $Ca^{2+}$ release from internal stores provide the slow variables that couple to spiking to produce irregular bursting near period-doubling cascades and related bifurcations. In particular, our results highlight the crucial role of SK channels and $Ca^{2+}$ dynamics, mediated by NMDA receptors and $Ca^{2+}$ release from the ER, towards shaping firing activity.

## A simplified phenomenological model of burst firing based on a modified Hindmarsh-Rose model

Thus far, we have demonstrated that our biophysically detailed model accurately reproduces the action potential shape and the *in vivo* temporal firing patterns of ELL pyramidal cells. While biophysical modeling provides valuable insights into the mechanisms underlying neuronal spiking activity, simpler phenomenological models often offer a better

framework for studying circuit-level neural dynamics and for understanding which combinations are most relevant [64–66]. We developed a new simplified model based on a modified Hindmarsh-Rose (HR) model to account for the diverse firing patterns exhibited by these cells (see Methods). Unlike the previously introduced biophysical model, which provides detailed representation of the underlying ionic currents and $Ca^{2+}$ dynamics in full details, the 4-dimensional modified HR model is phenomenological in nature, capable of emulating the key features of ELL pyramidal cell activity, including slow adaptation and intrinsic firing patterns, making it a computationally efficient model to study circuit dynamics of ELL pyramidal cells. By introducing adaptation through a slow variable ($z$) and adaptation current ($u$), the modified HR model accurately replicates irregular burst oscillations observed in the full biophysical model, as well as the broad range of firing patterns seen in experimental recordings as shown below.

In order to validate the model's ability to reproduce the diverse firing patterns and chaotic bursting dynamics of ELL pyramidal cells, we performed bifurcation analysis with respect to the applied current ($I_{app}$) (Fig 5A). For lower values of $I_{app}$, the model exhibits a branch of stable equilibria (solid orange), which corresponds to the resting potential of the cell. As $I_{app}$ increases, this branch loses stability at a subcritical Hopf bifurcation (HB1), forming a branch of unstable equilibria (dashed black). This unstable branch then folds twice at two saddle-node bifurcations (SN2 on the left and SN1 on the right; see bottom inset) before crossing another supercritical Hopf bifurcation (HB2), forming a stable branch of equilibria (solid orange) representing the depolarization block. At HB1, a branch of unstable periodic orbits (dashed blue) emerges and terminates at a homoclinic bifurcation (HM1; Fig 5A, top inset). Though non-physiological, a narrow regime of stable equilibria (solid orange) exists prior to the SN1, bounded on the left by a supercritical Hopf bifurcation (HB3) and on the right by a subcritical Hopf bifurcation (HB4) (Fig 5A, bottom inset). From HB3, a branch of stable periodic orbits (solid green) emerges and eventually undergoes period doubling bifurcation (PD1) that gives rise to chaotic small amplitude oscillations (dashed green; Fig 5A, bottom inset) before terminating at a homoclinic bifurcation (HM3). Similarly, a branch of unstable periodic orbits (dashed blue) also emerges from HB4 and terminates at a Hopf bifurcation (HB5) on the left of SN1 (Fig 5A, bottom inset). Likewise, a branch of unstable limit cycles (dashed blue) also emerges from a subcritical Hopf bifurcation (HB6) on the right of SN2 and terminate at a homoclinic bifurcation (HM2; Fig 5A). Finally, the branch of stable periodic orbits (solid green) that emerges from HB2 merges with a branch of unstable periodic orbits at a saddle-node of periodics bifurcation (SNP1), which in turn merges with a branch of stable periodic orbits (solid green) at a saddle-node of periodics bifurcation (SNP2; Fig 5A). The latter branch initially undergoes torus bifurcation (TS) on the right and period doubling bifurcation (PD2) on the left, giving rise to chaotic dynamics (Fig 5B) responsible for the ghostbursting behavior; they eventually terminate on the left by a cascade of infinite bifurcations that become increasingly hard to discern (Fig 5A, bottom inset). As $I_{app}$ increases, the modified HR model undergoes a sequential transition from quiescence, to tonic firing (Fig 5C, left), and then to bursting activity. In particular, at PD2 the dynamics switch from tonic spiking on the right of PD2 to ghostbursting with a variable number of spikes per burst on the left (Fig 5C, middle and right), in a manner consistent with the biophysical model (S4B Fig). This sequence of transitions underlies the emergence of irregular burst oscillations. The continuation analysis reaching from PD2 into the chaotic bursting regime on the left reveals multistability, eventually terminating into the chaotic attractor regime (rather than a homoclinic bifurcation). Furthermore, for values of $I_{app}$ below the threshold for spiking originating from HB1, the model exhibits sensitivity to initial conditions, producing either no spiking or single spiking events before returning to the stable resting state reminiscent of type IV excitability (S5 Fig).

Overall, these results illustrate the rich dynamical repertoire of the modified HR model through Hopf, period-doubling, and saddle-node of periodics bifurcations. They highlight the ability of the model to generate ghostbursting reminiscent of irregular burst oscillations observed in ELL pyramidal cells. One thus would expect that, with noise, the model can transition between stable equilibrium, tonic spiking, and chaotic ghostbursting patterns (Fig 5B), allowing the model to generate diverse activity patterns that mimic the *in vivo* ELL pyramidal cell activity.

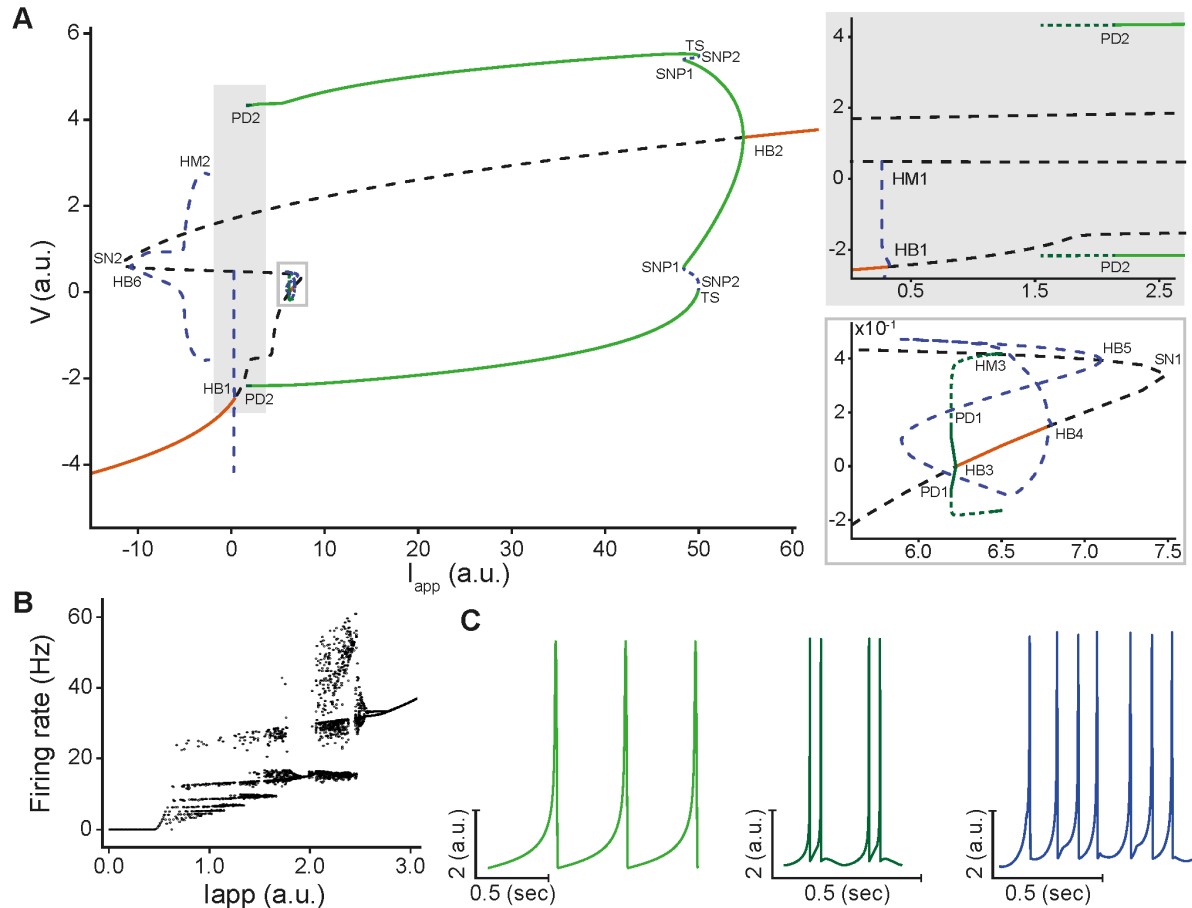

**Fig 5. Bifurcation analysis of the deterministic modified Hindmarsh-Rose (HR) model shows similar dynamic regimes as those of the full model. (A)** One-parameter bifurcation diagram of the voltage variable (V) of the modified HR model with respect to the applied current ($I_{app}$), illustrating the various regimes of behavior corresponding to branches of stable equilibria (orange line), branches of unstable equilibria (dashed black line), branches of stable periodic orbits (green lines), branches of bursting orbits (dashed green line), and branches of unstable periodic orbits (dashed blue line). Within the range of $I_{app}$ considered, the model undergoes several types of bifurcation points, including 2 saddle-node bifurcations (SN1, SN2), 6 Hopf bifurcations (HB1-HB6), 2 saddle-node bifurcations of periodic orbits (SNP1, SNP2), 2 homoclinic bifurcations (HM1, HM2), 2 period-doubling bifurcations (PD1, PD2) and 1 torus bifurcation (TS), some of which define the boundaries of the various regimes of behavior identified. The two insets provide magnified views of the areas enclosed by the bounding boxes in the main figure: the top inset corresponds to the larger, more physiologically relevant box, while the bottom one corresponds to the smaller, unphysiological box. **(B)** Plot of the firing rate of the modified HR model over a 20 sec interval with respect to $I_{app}$, highlighting chaotic dynamics consistent with the biophysical model (compare to S4B Fig). **(C)** A sample of three principal firing patterns: tonic spiking ($I_{app} = 0.4$, left), ghostbursting in the form of doublets ($I_{app} = 1.9$, middle) and chaotic ghostbursting ($I_{app} = 1.1$, right). These patterns demonstrate the rich repertoire of activity captured by the modified HR model.

## Stochastic simulation of the modified HR model accurately reproduces ELL pyramidal cell spiking activity seen *in vivo*

Now that we have demonstrated the ability of the modified HR model to reproduce burst firing dynamics comparable to those observed in the full biophysical model, it remains to check if incorporating stochastic synaptic bombardment can generate the spiking activity of pyramidal cells observed *in vivo*. To do so, we have added an extracellular noise term representative of random excitatory and inhibitory inputs to the modified HR model. We then systematically varied its parameters to match the spike train features we obtained from intracellular recordings from ELL pyramidal cells.

Simulating the membrane potential of this example cell model (Fig 6A, top, teal), alongside the two adaptation variables, $z(t)$ (middle, red) and $u(t)$ (bottom, orange) that were introduced to capture the slower timescales of burst onset and termination, showed that, despite the model's phenomenological nature, it can reproduce the key features of the spiking dynamics of pyramidal cells observed experimentally; that includes the alternation between burst episodes and isolated spiking.

To assess how well the modified HR model captures variability across different cells, we simulated multiple parameter sets corresponding to distinct ELL pyramidal cells. The raster plots of the resulting spike trains for five representative model cells obtained by varying parameter values in the model (Fig 6B; see Methods) revealed a wide range of firing patterns from predominantly bursting to more isolated spiking, similar to observed *in vivo* activity. The stochastic modified HR model reproduced experimental firing statistics when parameters were optimized to minimize our multi-objective loss function (see Methods). Parameter sets with loss values <0.1 and non-significant Kolmogorov-Smirnov tests comparing interspike interval distributions ($p>0.05$) were considered successful fits. Under this criterion, the model reproduced the majority of experimental recordings, capturing both the bimodal ISI distributions and the observed mixture of burst and isolated spikes (Fig 6B). We then performed a statistical comparison between the experimental data and model simulations for key spike train metrics, including the mean firing rate ($mean_{fr}$), the burst fraction, and the mean ($mean_{isi}$), median ($median_{isi}$),

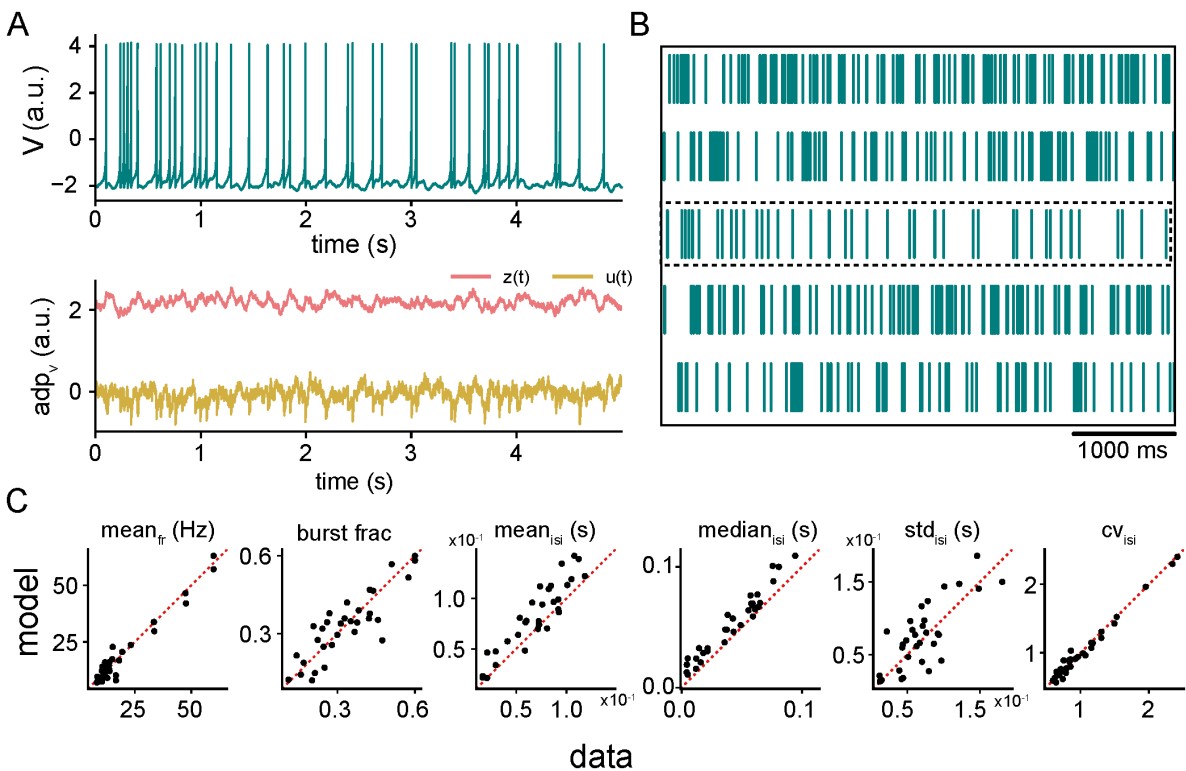

**Fig 6. Impact of stochastic synaptic input on the deterministic dynamics of the modified Hindmarsh-Rose (HR) model. (A)** Simulation of the voltage variable ($V$) of the modified HR model (top, teal), along with the slow adaptation variables $z$ (red) and $u$ (yellow), for an example ELL pyramidal cell receiving stochastic synaptic input (see Methods). **(B)** Raster spike trains obtained from simulations of the modified HR model for five different representing cells receiving stochastic synaptic input. These spike trains demonstrate the diversity of firing patterns across cells. **(C)** Statistical comparisons between experimental data (dark blue) and model simulations (teal) for spike train features of all recorded ELL pyramidal cells. From left to right: mean firing rate ($mean_{fr}$; $r = 0.97$, $p = 2.4 \times 10^{-21}$), burst fraction (burst frac; $r = 0.86$, $p = 2.1 \times 10^{-10}$), mean ($mean_{isi}$; $r = 0.91$, $p = 2.6 \times 10^{-13}$), median ($median_{isi}$; $r = 0.97$, $p = 1.9 \times 10^{-20}$), standard deviation ($std_{isi}$; $r = 0.77$, $p = 2.3 \times 10^{-07}$) and the coefficient of variation ($CV_{isi}$; $r = 0.98$, $p = 4.4 \times 10^{-17}$) of ISIs, respectively.

standard deviation ($std_{isi}$), and coefficient of variations ($CV_{isi}$) of the interspike intervals (Fig 6C). The scatter plots show a strong correspondence between the model simulations and the experimental data across all metrics (mean$_{fr}$ $r = 0.97$, $p = 2.4 \times 10^{-21}$; burst fraction $r = 0.86$, $p = 2.1 \times 10^{-10}$; mean$_{isi}$ $r = 0.91$, $p = 2.6 \times 10^{-13}$; median$_{isi}$ $r = 0.97$, $p = 1.9 \times 10^{-20}$; std$_{isi}$ $r = 0.77$, $p = 2.3 \times 10^{-07}$; and the $CV_{isi}$ $r = 0.98$, $p = 4.4 \times 10^{-17}$ of ISIs, respectively), demonstrating that the optimized model captures experimental spiking statistics with high fidelity.

In summary, the modified HR model provides a parsimonious yet effective framework for capturing the complex firing behaviors of ELL pyramidal cells *in vivo*. By combining a minimal set of adaptation variables with stochastic synaptic input, the modified HR model reproduces the principal burst dynamics and spiking variability documented experimentally, thereby offering a useful tool for future studies of network-level interactions and higher-order processing in the electrosensory system.

### Parameter sensitivity analysis highlights the importance of adaptation variables and stochastic noise in spiking dynamics

To investigate how the intrinsic parameters of the modified HR model contribute to the observed spiking activity, we conducted a parameter sensitivity analysis using Sobol indices (see Methods). Violin plots were used to visualize parameter distributions, where higher Sobol indices indicate greater influence on model dynamics (Fig 7A). This analysis identified the top parameters influencing model behavior, with the inhibitory coefficient of the slow adaptation variable ($\gamma_d$) and stochastic noise intensity of the secondary adaptation variable ($\sigma_u$) showing the highest sensitivity indices. Specifically, we found that $\gamma_d$ contributes to the transition between tonic and burst firing modes, while $\sigma_u$ introduces variability that closely aligns the model with *in vivo* spiking patterns. These results suggest that the interplay between intrinsic slow

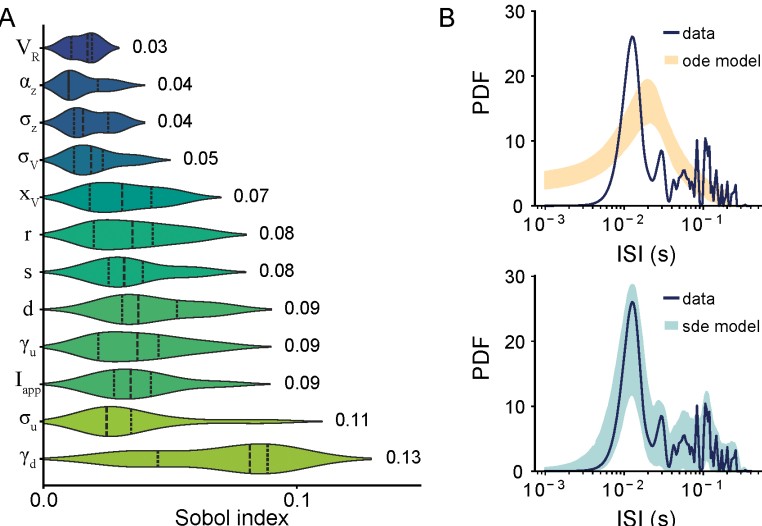

**Fig 7. Parameter sensitivity analysis of the modified Hindmarsh-Rose (HR) model reveals significant contribution of the slow adaptation variable and stochastic background noise in aligning it with *in vivo* spiking activity of ELL pyramidal cells. (A)** Sobol sensitivity indices for the top 10 model parameters in the modified HR model receiving stochastic synaptic input. The violin plots represent the distributions of parameter values, scaled by their importance as determined by the Sobol sensitivity indices for each parameter. The most sensitive parameters, based on higher Sobol indices, are $\gamma_d$ and $\sigma_u$, indicating the greater influence of the slow adaptation variable on model behavior compared to other parameters. **(B)** Comparison of the interspike interval (ISI) distributions between an example recorded ELL pyramidal cell (blue) and ISIs from model simulations without (orange, top) and with (teal, bottom) stochastic synaptic input, highlighting the significantly improved alignment of the model with experimental data in the presence of stochastic synaptic input (Kolmogorov-Smirnov test vs data: $D_{\text{ode model}} = 0.46$, $p_{\text{ode model}} = 0.0036$; $D_{\text{sde model}} = 0.146$, $p_{\text{sde model}} = 0.778$).

adaptation dynamics and stochastic extrinsic input is critical for reproducing the burst firing patterns and variability seen in experimental data.

Finally, to assess the role of the deterministic (i.e., intrinsic) components of the model in reproducing the temporal dynamics of spiking activity patterns observed *in vivo*, and their interaction with stochastic synaptic input, we fixed all parameters of the deterministic modified HR model based on the same example ELL pyramidal cell used earlier and simulated its activity with and without stochastic synaptic noise (Fig 7B). In the absence of stochastic input (top, orange), the model does not capture the burst firing dynamics seen experimentally, as quantified by significant differences between the ISI distributions obtained from the model and from our experimental data (Kolmogorov-Smirnov test vs data: $D_{\text{ode model}} = 0.46$, $p_{\text{ode model}} = 0.0036$). However, in the presence of stochastic background input (bottom, teal), the ISI distribution obtained from the model aligns closely with that of the recorded cell (shaded dark blue; Kolmogorov-Smirnov test vs data: $D_{\text{sde model}} = 0.146$, $p_{\text{sde model}} = 0.778$).

To further evaluate the importance of model structure, we simulated the classic HR model using the same parameters as in the modified version. The classic HR model produces an ISI distribution that differs significantly from experimental data (S6 Fig; Kolmogorov-Smirnov test vs data: $D_{\text{HR}} = 0.383$, $p_{\text{HR}} = 0.0065$). Specifically, it is shifted toward longer intervals (i.e., rightward displacement of the distribution), fails to capture fast bursting events, and lacks the characteristic bimodality observed *in vivo*. Additionally, the distribution is noticeably narrower, indicating reduced variability. These results highlight that the modifications introduced in our model, beyond just parameter tuning, are essential for accurately reproducing the complex temporal structure of ELL pyramidal cell spiking. These findings underscore the critical role of stochastic components in modeling synaptic bombardment and the high-conductance states characteristic of *in vivo* conditions.

## Discussion

### Summary of results

Neurons *in vivo* operate under different conditions than *in vitro* as they are constantly bombarded by synaptic inputs as well as neuromodulators that are typically absent *in vitro* [24,67,68]. Understanding how these factors interact with the intrinsic neuronal mechanisms is crucial for uncovering the critical computations performed by neural circuits in the behaving animal. In the present study, we addressed this gap by investigating how *in vivo* conditions affect the intrinsic burst firing displayed by ELL pyramidal cells *in vitro*. To do so, we developed a two-compartment biophysical model that combines the membrane voltage dynamics with $Ca^{2+}$ mobilization. After fitting the model parameters against intracellular recordings, we first demonstrated that the model successfully captures the intrinsic and extrinsic mechanisms underlying the firing patterns of these cells. Next, we used bifurcation analysis to identify how specific conductances, such as SK ($g_{\text{SK}}$) and NMDA ($g_{\text{NMDA}}$), shape transitions between quiescent, tonic, and bursting regimes, revealing complex multistable dynamics. We then designed a simplified model based on the modified Hindmarsh-Rose (HR) model to phenomenologically reproduce the firing patterns of ELL pyramidal cells seen *in vivo*. This model retained key dynamical features, including chaotic bursting, while offering computational efficiency, providing a robust framework for exploring the spiking activity of these cells. Finally, the parameter sensitivity analysis highlighted the critical role of slow adaptation mechanisms and stochastic input in shaping spiking patterns detected *in vivo*. The two models together offer a mechanistically grounded framework for understanding how intrinsic dynamics and *in vivo* conditions interact to shape the activity of ELL pyramidal cells.

Importantly, our use of both a biophysical and a phenomenological model provides complementary perspectives. The detailed biophysical model highlights the contributions of specific ionic currents (e.g., SK, NMDA) and $Ca^{2+}$ mobilization, thereby uncovering mechanistic substrates of burst dynamics. By contrast, the modified HR model abstracts these details while retaining the essential repertoire of firing regimes with much greater computational efficiency, enabling its application to large-scale network modeling studies. Despite some differences in their bifurcation structures, both models

reproduce the sequential transitions between quiescence, tonic firing, and bursting, as well as irregular dynamics near bifurcation cascades. The main limitation of the biophysical model is its computational cost, whereas the modified HR model sacrifices mechanistic specificity. Taken together, the convergence of these approaches strengthens our conclusions, demonstrating that the observed *in vivo* burst dynamics are robust across different levels of abstraction.

Interestingly, our modeling framework helped to explain the sequential transition from quiescence to tonic firing and then to bursting as current input increased, consistent with previous *in vitro* studies of ELL pyramidal cells [27,38,39]. Similarly, *in vivo* experiments have shown that stepwise changes in the EOD amplitude modulations initially elevate firing rates before adaptation reduces them, suggesting that alterations in synaptic drive can also shift cells between firing regimes [55,69].

**The impact of dendritic integration and stochastic synaptic input on burst firing**

Seminal *in vitro* studies of ELL pyramidal cells described a "ping-pong" mechanism driven by the backpropagation of somatic action potentials into the dendrites. This process leads to dendritic spikes and subsequent somatic depolarizing afterpotentials (DAPs) that trigger further spikes until they eventually terminate due to failure of dendritic spike backpropagation [37–39,41]. Such somato-dendritic interaction in ELL pyramidal cells was computationally described as "ghost-bursting" [39,41]. In this regime, periodic-orbit bifurcations couple the fast somatic and slower dendritic dynamics, with burst termination occurring when trajectories are reinjected near the ghost of a saddle–node bifurcation of fixed points, giving rise to long and irregular inter-burst intervals. The resulting bursts are typically short and variable in spike count, with doublets representing a special case. While saddle–node and period–doubling bifurcations underlie this mechanism when the dynamics are analyzed with respect to applied current, these bifurcations are not unique to ghostbursting and also occur in other neuronal bursting models [70,71]. In contrast, *in vivo* recordings, including those presented here, consistently show bursts that are typically shorter, often lack prominent DAPs or pronounced afterhyperpolarizations (AHPs), and exhibit more variable interspike intervals (ISIs) [12,25,43,54]. Our biophysical modeling provides a novel mechanistic insight to explain these discrepancies by incorporating key features unique to *in vivo* conditions. Specifically, the model demonstrates that dendritic $Ca^{2+}$ influx, primarily mediated by NMDA receptor dynamics, activates SK channels whose hyperpolarizing effects act to prematurely terminate bursts before the full development of the ping-pong mechanism [25]. This truncation shortens burst duration and reduces DAP prominence, resulting in burst dynamics that closely align with those observed *in vivo*. The irregular (i.e., chaotic) bursting regimes observed in our model emerge when slow intracellular $Ca^{2+}$ dynamics interact with the fast spiking subsystem through fast–slow interactions. This behavior is consistent with previous $Ca^{2+}$-dependent models of chaotic oscillations in bursting neurons, which showed that such interactions between slow $Ca^{2+}$ processes and fast membrane excitability can generate chaotic activity [63]. In our model, the slow variables are provided by NMDA-mediated $Ca^{2+}$ influx and release of $Ca^{2+}$ from internal stores, allowing SK channels to shape burst firing and delimit regime boundaries. Thus, while the underlying molecular substrates differ from other systems where $IP_3$-dependent $Ca^{2+}$ signaling is prominent, the general mechanism involving slow $Ca^{2+}$ processes interacting with fast spiking appears to be a robust route to chaotic bursting across neuronal types. Indeed, this result is consistent with experimental studies showing that pharmacological inhibition of SK channels by apamin [46], or blockade of intracellular $Ca^{2+}$ using BAPTA [25,47], restores long-duration bursts with prominent DAPs *in vivo* by preventing the premature termination of bursts. Complementing this gating mechanism is the influence of extrinsic synaptic input, which introduces continuous membrane potential variability. This variability contributes to irregular burst timing and disrupts the structured burst patterns observed *in vitro*, which otherwise exhibit strong correlations with specific stimulus properties [9,26,72]. By applying parameter sensitivity analysis, we further highlighted in this study the importance of these factors by revealing that both slow adaptation dynamics and stochastic noise intensity are dominant determinants of the observed variability and structure of *in vivo* burst events, suggesting a crucial interplay between intrinsic cellular properties and the presence of stochasticity caused by active synaptic bombardment. It is worth noting that while we

modeled synaptic bombardment as additive current noise, more detailed conductance-based approaches yield multiplicative, state-dependent noise [9,17]. Additive noise has nonetheless been widely used to approximate *in vivo*–like synaptic fluctuations and reproduces realistic firing variability across different neuron types [18,73–75], supporting our modeling approach. With this choice of noise, the dynamic interplay between stochastic fluctuations and slow adaptation emerged as a key factor in distinguishing *in vivo* burst firing from those patterns observed *in vitro*.

## Modulation of firing dynamics by feedback and neuromodulator serotonin

It is well-known that ELL pyramidal cells *in vivo* are subject to substantial neuromodulatory [45,76,77] and synaptic input [35,43], which can dynamically influence their spiking activity and burst firing patterns over time. Within the ELL, neuromodulatory inputs, including acetylcholine [78,79], and serotonin [80–82] are known to broadly regulate neuronal function by modifying excitability and responsiveness to stimulation. Our models provide a robust framework for understanding how such factors modulate firing dynamics in ELL pyramidal cells by exploiting the cell's intrinsic multistability. For instance, it has been shown experimentally that serotonin released from raphe nuclei inhibits SK channels, reducing the medium afterhyperpolarization and promoting burst firing in ELL pyramidal cells [27]. SK channels have also been shown to play a critical role in regulating neuronal excitability and frequency tuning [46]. Our model builds on these findings by quantitatively predicting that reducing SK conductance ($g_{SK}$) lowers the burst threshold and expands the bursting regime [12,21,45,82]. The model can further be used to explore how serotonin affects membrane excitability and promotes burst firing, as seen experimentally, by leveraging the intrinsic dynamics (e.g., SK channels), thereby offering mechanistic insight into experimental results showing serotonin modulation enhances the encoding of behaviorally relevant stimuli such as specific communication signals [45,76]. More generally, our bifurcation analyses demonstrated that changes in synaptic input or neuromodulatory state can shift the boundaries between quiescent, tonic, and bursting regimes. This implies that a single ELL pyramidal cell can switch between these regimes depending on its external inputs: remaining quiescent under one condition but firing tonically under another, or exhibiting burst firing when SK conductance is reduced. Both the detailed biophysical and the modified HR models capture this flexibility, underscoring how synaptic and neuromodulatory modulation can dynamically reconfigure the firing patterns of individual cells.

In addition to neuromodulation, *in vivo* spiking activity of neurons is significantly affected by feedback from higher brain areas, which can reflect behavioral state, attention, or predictive processing [83,84]. Such feedback can dynamically regulate ELL pyramidal cell excitability and responsiveness to sensory input [25,37,43] through both stochastic synaptic bombardment [26] and NMDA receptor-mediated modulation of membrane potentials [69,85,86]. Our bifurcation analysis demonstrates that changes in NMDA conductance ($g_{NMDA}$) strongly influences the boundaries between firing regimes, thereby offering a dynamical basis for the established ability of feedback pathways to control whether a neuron operates in silent, tonic, or bursting modes [43]. Even brief or subtle shifts in feedback or neuromodulatory factors, according to our model, can shift cells across these regime boundaries, consistent with experimental observations [45,76,82]. Furthermore, the continuous presence of stochastic synaptic activity in *in vivo* introduces additional membrane potential variability and can enhance the responsiveness of pyramidal cells to changes in feedback and neuromodulation [9,10,26]. Together, these mechanisms provide a flexible means by which the firing mode of ELL pyramidal cells can be rapidly and reversibly modulated via neuromodulation [45,48,78,82], feedback strength [23,43,85], and ongoing synaptic activity [26,35]. Our model thus provides a biophysical basis for the observed dynamic control of burst firing *in vivo* and explains how physiological modulation can drive transitions between firing states.

## Multistability: A mechanism for adaptive neuronal computation

A key finding from our bifurcation analysis is the identification of complex dynamical behaviors characterized by multistability in both models. The biophysical model exhibits multistability near the saddle-node (SN1) and period doubling (PD1) bifurcations, where quiescent and bursting states coexist under the same depolarizing current [87,88]. Such sensitivity to

initial conditions could have biological relevance. Experimental studies in other systems have also demonstrated neuronal bistability, such as dendritic $Ca^{2+}$ dynamics in Purkinje cells [89] and in neocortical pyramidal neurons driven by persistent synaptic input [14], supporting the notion that multiple stable states can be accessed depending on prior activity and perturbations. *In vivo*, ongoing synaptic input and intrinsic noise provide perturbations that move neurons between coexisting states, thereby enabling flexible switching between quiescent, tonic and bursting regimes. Furthermore, the chaotic bursting regime itself exhibits multiple coexisting bursting patterns (multistability within bursting), reflecting the complex attractor structures that are characteristic of neurons with slow $Ca^{2+}$-related dynamics [87,90]. Similarly, the modified HR model displays multistable bursting dynamics (near PD2), further illustrating the model's capability to reproduce the complex firing dynamics observed in the biophysical model. This multistability has significant implications for neuronal coding [70,91,92], facilitating, for example, rapid and context-dependent switching between different firing modes, such as quiescent, tonic spiking, and bursting, without requiring long-term synaptic changes. Despite the stochastic nature of synaptic inputs, the intrinsic properties of dendritic and axonal compartments enable neurons to generate and differentiate between single spikes and bursts, assigning distinct coding roles to each firing mode [93–95]. While these coding roles arise from the compartmental integration and active properties of neurons, synaptic noise can act as a perturbation that pushes the neurons across firing regime boundaries within its multistable landscape, even though it doesn't directly encode stimulus features [11,21,41,96]. For instance, in the subthreshold stable node regime of both models, a transient perturbation can elicit a single spike consistent with Type IV excitability [61,62]. The mechanism typically involves the neuron being poised near a bifurcation (such as a SNIC or a homoclinic bifurcation), where a brief input allows a single spike excursion before returning to the stable resting state. Such single-spike excitability can be leveraged as a precise temporal coding mechanism, enabling neurons to transmit information with high temporal fidelity by signaling the exact onset of a stimulus or encoding fine temporal features through isolated spikes [61]. Indeed, this capability arises from the rapid and temporally precise nature of single spikes, allowing them to encode fine temporal patterns that are critical for sensory discrimination [54,97,98]. In contrast, bursts offer a robust means of signaling in noisy environments and may encode not only more generalized or invariant stimulus attributes [4,12,26,44], but also different time scales [70]. Taken together, this complex dynamical landscape, marked by the coexistence of multiple firing states, can enable neurons to dynamically adjust their firing mode in response to transient inputs or ongoing synaptic fluctuations. Such multistability thus serves as a powerful mechanism for adaptive computation, allowing neurons to flexibly alter their coding strategies in accordance with changing sensory contexts or behavioral demands [3]. At the same time, multistability may also underlie dysfunctional regimes when switching between coexisting attractors leads to irregular pathological activity, as reported previously in other model neurons [88].

## Generalizability and implications for other systems

The dual-modeling approach employed in this study, using a detailed biophysical model for mechanistic grounding alongside a simplified phenomenological model, can be applied to other cell types to bridge the gap between *in vitro* and *in vivo* dynamics [9,10,24,67,99]. This integrated framework allows direct testing of specific ionic mechanisms (e.g., SK channel roles), while at the same time providing a more compact alternative to the biophysical model that can reproduce the principal firing patterns of ELL pyramidal cells. The mechanisms identified here for *in vivo* spiking activity and burst dynamics likely extend beyond the electrosensory system. While burst firing is a widespread phenomenon across various brain regions [4,100], the electrosensory system shares notable similarities with other sensory modalities, including the mammalian vestibular [101,102], visual [103], and auditory [104] systems. Understanding how specific *in vivo* mechanisms, such as the $Ca^{2+}$-related dynamic interactions identified in this study, shape the structure and function of bursts is therefore broadly relevant. Moreover, the multistability uncovered by our models suggests a general computational principle: that dynamic switching between distinct firing modes may support adaptive coding across sensory, motor, and cognitive domains [3,105]. Finally, our modeling framework also enables broader applications, such as probing cellular

heterogeneity [21,106], incorporating activity-dependent plasticity [23,35], and leveraging the computational efficiency of the modified HR model for large-scale simulations [107,108].

## Materials and methods

### Ethics statement

All procedures involving animals in this study were reviewed and approved by the McGill University Animal Care Committee (#5285) and were conducted in accordance with the regulations set by the Canadian Council on Animal Care.

### Experimental setup

The wave-type weakly electric fish *Apteronotus leptorhynchus* ($N = 8$) of either sex was used exclusively in this study. The fish were sourced from commercial tropical fish suppliers and housed in groups ranging from 2 to 10 individuals. Water conditions were carefully maintained at temperatures between 26 and 29 °C, with conductivities ranging from 300 to 800 $\mu$S·cm$^{-1}$, following established care guidelines [109].

### Surgery

Details of the surgical procedures have been previously described in detail [25,104,110]. Briefly, animals were immobilized for electrophysiological recordings via intramuscular administration of tubocurarine (0.1-0.5 mg, Sigma). Fish were then placed in an experimental tank (30 cm × 30 cm × 10 cm) filled with water from their home tank and provided with a constant flow of oxygenated water through the mouth at 10 mL/min rate. Subsequently, the animal's head was locally anesthetized with lidocaine ointment (5%; AstraZeneca, Mississauga, ON, Canada), the skull was partly exposed, and a small window was opened over the hindbrain.

### Recordings

Intracellular recordings were conducted *in vivo* from ELL pyramidal cells of either cell type (ON- or OFF-) using sharp electrodes, following standard procedures [25,45,55]. Recordings were made under baseline conditions, defined by the presence of the animal's continuous but unmodulated EOD. Indeed, the EOD acts only as a carrier signal and does not constitute sensory input unless perturbed [30,111]. To eliminate such perturbations, animals were kept individually in isolated tanks without objects, conspecifics, or external electric fields as done previously [21,25,45]. Thus, the recorded activity reflects intrinsic and synaptic mechanisms in the absence of external sensory drive. During recording, micropipettes had resistances ranging from 20 to 80 M$\Omega$ and were filled with 3 M potassium chloride (KCl), as done previously [112]. Pyramidal cells were recorded at depths ranging from 300 to 800 $\mu$m below the brain surface, corresponding to the centrolateral and lateral ELL segments based on established anatomical landmarks [113,114]. Recordings were digitized at 10 kHz using CED 1401 plus hardware and Spike2 software (Cambridge Electronic Design, Cambridge, UK) and stored on a hard drive for further analysis. Overall, we recorded from a total of $n = 32$ ELL pyramidal cells across $N = 8$ fish. Individual intracellular recordings lasted 5 sec, and all spike train statistics (e.g., mean firing rate, burst fraction, ISI distributions) were computed across the entire recording period. The number of cells per fish ranged from 1 to 10. To assess the relative contributions of within- versus between-animal variability in electrophysiological features, we quantified action potential characteristics (threshold, upstroke, peak, downstroke, trough, ADP) separately for each fish. We then computed intra-class correlation coefficients (ICC) using a two-way random-effects model with absolute agreement to partition the variance of each feature into between-fish and within-fish components. The resulting ICC values indicated that most of the variability arose within fish rather than across fish (ICC$_{threshold}$ = 0.33; ICC$_{upstroke}$ = 0.04; ICC$_{peak}$ = 0.31; ICC$_{downstroke}$ = 0.39; ICC$_{trough}$ = 0.26; ICC$_{ADP}$ = 0.49), consistent with previous studies showing substantial heterogeneity among superficial, intermediate, and deep ELL pyramidal cells in their intrinsic conductances and synaptic feedback [12, 30,43,111].

## Computational model

**Biophysical model of ELL pyramidal cells.** We developed a two-compartmental biophysical model of ELL pyramidal cell activity [21,25,39]. The model consists of two compartments and is based on the Hodgkin-Huxley formalism. The equations describing the membrane voltages in the somatic ($V_S$) and dendritic ($V_D$) compartments are given by

$$C_m \frac{dV_S}{dt} = I_{app} - I_{Na,S} - I_{K,S} - \frac{g_c}{\kappa}(V_S - V_D) - I_{leak,S} \tag{1}$$

$$C_m \frac{dV_D}{dt} = I_{syn} - I_{Na,D} - I_{K,D} - I_{SK} - I_{NMDA} - \frac{g_c}{1-\kappa}(V_D - V_S) - I_{leak,D} \tag{2}$$

where $C_m$ is the membrane capacitance, $I_{Na,i}$ and $I_{K,i}$ are the fast inward Na$^+$ and the slow outward delayed rectifier K$^+$ currents in the soma ($i = S$) and the dendrite ($i = D$), respectively, $I_{app}$ is the depolarizing current to the somatic compartment, $I_{syn}$ is the synaptic current to the dendritic compartment, $I_{SK}$ and $I_{NMDA}$ are the outward Ca$^{2+}$-dependent small-conductance K$^+$ and the inward NMDA Ca$^{2+}$ currents acting on the dendrite, respectively, and $I_{leak,S}$ and $I_{leak,D}$ are the passive leak currents. The two compartments are linked together through a resistor, where $g_c$ is the maximum conductance and $\kappa$ is the somatic-to-dendritic area ratio. The currents $I_{Na,i}$ and $I_{K,i}$ ($i = S, D$) are necessary to generate somatic action potentials and the proper spike backpropagation that yields somatic depolarizing afterpotentials (DAPs) [39]. The two currents $I_{NMDA}$ and $I_{SK}$ were added to the dendritic compartment because of the important role they play in regulating spiking activities of ELL pyramidal cells both *in vitro* [46] and *in vivo* [25,115,116]. The kinetics of the ionic currents included in each compartment ($i = S, D$) are given by

$$I_{Na,S} = g_{Na,S} m_{\infty,S}^2 (1 - n_S)(V_S - V_{Na}) \tag{3}$$

$$I_{K,S} = g_{K,S} n_S^2 (V_S - V_K) \tag{4}$$

$$I_{leak,S} = g_{leak}(V_S - V_{leak}) \tag{5}$$

$$I_{Na,D} = g_{Na,D} m_{\infty,D}^2 h_D (V_D - V_{Na}) \tag{6}$$

$$I_{K,D} = g_{K,D} n_D^2 p_D (V_D - V_K) \tag{7}$$

$$I_{SK} = g_{SK} \frac{[Ca^{2+}]_i^2}{[Ca^{2+}]_i^2 + k_{Ca}^2}(V_D - V_K) \tag{8}$$

$$I_{NMDA} = g_{NMDA} B(V_D)[O](V_D - V_{Ca}) \tag{9}$$

$$I_{leak,D} = g_{leak}(V_D - V_{leak}) \tag{10}$$

where $g_{j,i}$ ($j$ = Na, K, leak, SK, NMDA and $i = S, D$) are the maximum conductances, $V_j$ ($j$ = Na, K, Ca, leak) are the Nernst potentials, $[Ca^{2+}]_i$ is the Ca$^{2+}$ concentration in the dendritic compartment ($\mu$M), $k_{Ca}$ is the half-maximum activation of the SK channel ($\mu$M), $B(V_D)$ is the magnesium block [56]. It should be noted that in our formulation, $I_{NMDA}$ denotes specifically the Ca$^{2+}$ component of the mixed NMDA conductance, as this is the current that activates SK channels and shapes burst termination. Accordingly, we used ($V_D - V_{Ca}$) as the driving force, considering only the Ca$^{2+}$ contribution and omitting the Na$^+$/K$^+$ components that shift the mixed NMDA current reversal toward 0 mV. The equation for $B(V_D)$ is given by

$$B(V_D) = \frac{1}{1 + \exp(-0.062 V_D)[Mg^{2+}]_o / 3.57} \tag{11}$$

where $[Mg^{2+}]_o$ is the extracellular magnesium concentration (mM), $[O]$ is the open probability of the NMDA receptors defined by a Markov model comprised of three closed, one open and one desensitized states with transition rates

$I_i$ ($i = b, u, c, o, d, r$) between them, $m_{\infty,i}$ ($i = S, D$) is the steady state activation of $I_{Na,i}$, and $x$ ($x = n_S, n_D, h_D, p_D$) are the activation/inactivation gating variables whose steady states and time constants are denoted by $x_\infty$ and $\tau_x$, respectively, and whose dynamics are governed by the equation

$$\frac{dx}{dt} = \frac{x_\infty - x}{\tau_x}. \tag{12}$$

The steady state activation/inactivation functions are given by

$$x_\infty(V_i) = \frac{1}{1 + \exp(-(V_i - V_{x_i})/s_{x_i})}; \quad x_i = m_i, n_i, h, p; \quad i = S, D. \tag{13}$$

In this model formalism, it was assumed that $I_{SK}$ and $I_{NMDA}$ are co-localized within spines. We therefore did not consider $Ca^{2+}$ diffusion within/between spines.

The NMDA receptor activation depends on the concentration of glutamate following principles of ligand-gated channels described previously [56], where the transition rates between unbound and bound states are based on the concentration of ligand. In our model, the timing of the release of glutamate follows a Poisson process with firing rate $\lambda_{glu}$. The glutamate concentration following each release event was described by an alpha function [56], given by

$$[glu](t) = A_{glu}(t - \tau_r)e^{-\frac{t - \tau_r}{\tau_d}} H(t - \tau_r) \tag{14}$$

where $\tau_r$ is the timing of glutamate release, determined by presynaptic spike times, $\tau_d$ is the time constant of the alpha function, $A_{glu}$ is a scaling constant and $H(t - \tau_r)$ is the Heaviside step function. The parameter $\tau_d$ was chosen to be fast enough to prevent glutamate release events from overlapping.

The $Ca^{2+}$ mobilization model followed the flux-balance formalism; it describes fluxes across the cell and ER membranes in the dendritic compartment. $Ca^{2+}$ mobilization across the cell membrane includes three fluxes through the NMDA receptors: $J_{NMDA} = \alpha I_{NMDA}$, where $\alpha = \frac{1}{2F\bar{V}_D}$ ($F$ is Faraday's constant and $\bar{V}_D$ is the volume of the dendritic compartment), plasma membrane $Ca^{2+}$ ATPases (PMCA) pumps: $J_{PMCA}$, and leak: $J_{INLeak}$. $Ca^{2+}$ mobilization across the ER membrane, on the other hand, includes three fluxes through the IP3Rs: $J_{IP3R}$, sarco/endoplasmic reticulum ATPases (SERCA) pumps: $J_{SERCA}$, and leak: $J_{ERLeak}$. The Li-Rinzel model was adopted to describe IP3R kinetics [57,58,117]. It is given by

$$\frac{d[Ca^{2+}]_i}{dt} = f_c(J_{NMDA} - J_{PMCA} + J_{IP3R} + J_{ERleak} - J_{SERCA} - J_{INleak}) \tag{15}$$

$$\frac{d[Ca^{2+}]_{ER}}{dt} = f_{ER}\gamma(J_{SERCA} - J_{IP3R} - J_{ERleak}) \tag{16}$$

where $f_c$ ($f_{ER}$) is the fraction of free $Ca^{2+}$ concentration in the cytosolic (ER) component of the dendrite, and $\gamma$ is the volume ratio of cytosol to ER in the dendrite. The fluxes through the PMCA and SERCA pumps are given by

$$J_\eta = \nu_\eta \frac{[Ca^{2+}]_i^\iota}{[Ca^{2+}]_i^\iota + K_\eta^\iota}, \quad \eta = PMCA, SERCA \tag{17}$$

where $\iota = 2$ is the Hill coefficient, $\nu_\eta$ are the maximum flux rates ($\mu M/s$), and $K_\eta$ are the half-maximum activations for $Ca^{2+}$ flux ($\mu M$). Fluxes due to leak across the cell and ER membranes are given by

$$J_{INleak} = \nu_{INleak}([Ca^{2+}]_o - [Ca^{2+}]_i) \tag{18}$$

$$J_{ERleak} = \nu_{ERleak}([Ca^{2+}]_{ER} - [Ca^{2+}]_i) \tag{19}$$

where $\nu_\xi$ ($\xi$ = INleak, ERleak) are the maximum flux rates ($\mu$M/s). Finally, flux through IP3Rs is adopted from the Li-Rinzel model [57] and is given by

$$J_{\text{IP3R}} = \nu_{\text{IP3R}} m^3_{\infty,\text{IP3R}} n^3_{\infty,\text{IP3R}} h^3_{\text{IP3R}} ([Ca^{2+}]_{ER} - [Ca^{2+}]_i) \tag{20}$$

where $\nu_{\text{IP3R}}$ is the maximum flux rate ($\mu$M/s),

$$m_{\infty,\text{IP3R}} = \frac{[IP3]}{[IP3] + d_1} \tag{21}$$

$$n_{\infty,\text{IP3R}} = \frac{[Ca^{2+}]_i}{[Ca^{2+}]_i + d_5} \tag{22}$$

$$\frac{dh_{\text{IP3R}}}{dt} = \frac{h_{\infty,\text{IP3R}} - h_{\text{IP3R}}}{\tau_{h_{\text{IP3R}}}} \tag{23}$$

$$h_{\infty,\text{IP3R}} = \frac{Q_2}{Q_2 + [Ca^{2+}]_i} \tag{24}$$

$$\tau_{h_{\text{IP3R}}} = \frac{1}{a(Q_2 + [Ca^{2+}]_i)} \tag{25}$$

$$Q_2 = d_2 \frac{[IP3] + d_1}{[IP3] + d_3}. \tag{26}$$

Note that $[IP3]$ is the cytosolic concentration of IP3 in the dendrite ($\mu$M).

The synaptic input current, $I_{syn}$, was used to represent all the stochastic background excitatory and inhibitory pre-synaptic activity, defined by $W_x(t)$ ($x = E, I$), applied to the dendritic compartment [59,60]. It has been previously suggested that synaptic inputs have $1/f^\beta$ power spectra, where $\beta$ determines the exponent of steepness of the slope of the power spectrum [60,118]. Based on this, the total synaptic input incorporated into the model can be described by

$$I_{syn}(t) = \sum_{x \in \{E,I\}} \sigma_x W_x(t) = \sigma_\eta W_\eta(t) \tag{27}$$

where $\sigma_\eta$ is the total noise intensity obtained by integrating the two components, $W_\eta(t)$ is the sum of both excitatory and inhibitory pre-synaptic inputs reconstructed using the following expression

$$W_\eta(t) = \sum_\omega \sqrt{S_x(\omega)} \cos(\omega - \phi(\omega)) \tag{28}$$

with $\omega = 2\pi f$ and $\phi(\omega) \in [0, 2\pi]$ is a random phase multiplied by the spectrum in the frequency domain. Once the spectrum is reconstructed, the synaptic time series can be obtained using the inverse Fourier transform of the spectrum as described previously [119].

**Modified Hindmarsh-Rose model.** The classical Hindmarsh-Rose (HR) model is a well-known three-dimensional system used to study neuronal spiking and bursting behaviors [64,66]. However, in its classical form, this model shares a common limitation with many other phenomenological bursting models in that the intra-burst spike frequency typically decreases over the course of the burst cycle. In contrast, ELL pyramidal cells exhibit an increasing spike frequency during the intra-burst interval. Furthermore, the classical HR model displays abrupt transitioning from a quiescent state to burst firing as the depolarization current is increased, whereas ELL pyramidal cells exhibit gradual transitions from quiescence to tonic firing before reaching the bursting state in response to increasing current. To address these discrepancies and better capture experimentally observed spiking and bursting patterns in ELL pyramidal cells *in vivo*, we developed a

simplified phenomenological model based on the Hindmarsh-Rose (HR) model. The classical HR model comprises a single fast variable ($V$) that mimics the membrane potential traces of neurons, a recovery variable ($y$) that controls the rapid recovery processes following action potentials, and a slow adaptation variable ($z$) that reflects the slow adaptation mechanisms that modulate neuronal excitability over longer timescales, regulating the spike frequency. We extended the HR model by introducing a new secondary slow variable ($u$) that modulates the spiking activity through an activation gating mechanism, enabling dynamic regulation of neuronal excitability based on membrane potential. The model is given in Itô SDE form as:

$$dV = \left[y - aV^3 + bV^2 - z - u + I_{app}\right] dt \; + \; \sigma_V \, dW_V(t), \tag{29}$$

$$dy = \left[c - dV^2 - y\right] dt \; + \; \sigma_y \, dW_y(t), \tag{30}$$

$$dz = \left[r\left(s(V - V_R) - z\right)\right] dt \; + \; \sigma_z \, dW_z(t), \tag{31}$$

$$du = \left[-\gamma_u \, n_{\infty,V}(V) + \alpha_z \, h_{\infty,z}(z) - \gamma_d \, u\right] dt \; + \; \sigma_u \, dW_u(t), \tag{32}$$

where $I_{app}$ is an external input current representing the stimulus input to the cell, and noise terms $\sigma_i$ ($i = V, y, z, u$) incorporate all stochastic fluctuations at each time step accounting for both intrinsic and external background noise in each equation [59,120]. Here, $W_i(t)$ are independent standard Wiener processes, and $dW_i(t)$ are their increments, satisfying $\mathbb{E}[dW_i] = 0$, $\mathrm{Var}[dW_i] = dt$, and $\mathbb{E}[dW_i dW_j] = 0$, for $i \neq j$. It should be noted that the modified HR model is non-dimensional with respect to the voltage- and current-like variables ($V, y, z, u, I_{app}$). By contrast, time was explicitly scaled to seconds in order to allow direct comparison of simulated firing statistics with experimental data, and firing frequency is therefore reported in Hz.

The dynamics of the newly introduced adaptation variable $u$ is governed by the interaction between voltage-dependent activation ($n_{\infty,V}$) and a secondary inactivation process ($h_{\infty,z}$). In this formalism, $n_{\infty,V}$ represents the steady-state activation function governed by membrane potential $V$, where

$$n_{\infty,V} = \frac{1}{2}\left(1 + \tanh\left(\frac{V - x_V}{k_n}\right)\right) \tag{33}$$

and $h_{\infty,z}$ represents the steady-state inactivation function dependent on the slow adaptation variable $z$,

$$h_{\infty,z} = \frac{1}{2}\left(1 + \tanh\left(\frac{z - x_z}{k_h}\right)\right), \tag{34}$$

where $x_V$ and $x_z$ denote the baseline values, and $k_n$, $k_h$ represent the slopes of these gating variables, determining how rapidly they respond to changes in $V$ and $z$, respectively.

This formulation allows the model to capture adaptive modulation of neuronal excitability, incorporating a dynamic activation-inactivation mechanism that extends the classical HR model. The inclusion of $u$ introduces additional complexity that enhances the model's ability to reproduce experimentally observed spiking and bursting patterns in ELL pyramidal cells.

## Data analysis

**Spike detection and feature extraction.** Both the data obtained from intracellular recordings and the model simulations were analyzed using a custom code built upon the Intrinsic Physiology Feature Extractor (IPFX) library (https://ipfx.readthedocs.io/) from Allen Brain Institute. Spike detection was performed using a threshold-based algorithm (-30 mV relative to baseline for the data and the ELL pyramidal cell model, and 0 mV for the modified HR model). To avoid artifacts, the maximum allowable time between consecutive spikes was set to 1 ms. Individual action potentials were first isolated

from voltage recordings by extracting segments of the membrane potential trace within a $\pm 4$ ms window around each spike peak. For each action potential, five AP features were extracted, including voltage threshold for spiking, peak amplitude (peak-to-trough voltage), trough amplitude, voltage at the midpoint of the upstroke phase, voltage at the midpoint of the downstroke phase (voltage at 50% amplitude of the upstroke and downstroke phases), and trough amplitude. In addition, the afterdepolarization potential amplitude ($adp_v$) was extracted following the Allen Institute's IPFX conventions. After detecting the spike's threshold, peak, and trough, the algorithm searched the post-spike segment for a local depolarizing maximum ('ADP'). The absolute membrane voltage at this point was recorded as $adp_v$. If no ADP maximum was detected (e.g., for the first spike in a burst where no clear ADP was visible), the algorithm returned NaN, and those spikes were excluded from ADP statistics. Spikes clipped at the boundaries of the analysis window were also excluded to avoid errors. These features together were used for comparison between the ELL pyramidal cell model and experimental data during optimization. To reduce variability across spikes, the average of each action potential feature across multiple spikes was computed and used for subsequent analyses.

**Inter-spike interval (ISI) distribution estimates.** To estimate ISI distributions, spike times were extracted following spike detection, based on the timing of detected peak amplitudes from either recordings or simulations (5 sec duration). ISIs were calculated as the differences between consecutive spike times. For visualization, histograms of ISI distributions were constructed using 2 ms bin widths. Additionally, kernel density estimation (KDE) was applied to generate a smoothed representation of the ISI distributions, using a Gaussian kernel with bandwidth selected according to Scott's rule [121]. To obtain error intervals around the ISI distributions, we used a bootstrapping approach with 50 resamples, where 4 sec segments of the spike time data were randomly subsampled. The variance around the mean KDE estimate across these samples was then calculated and plotted as the error interval.

**Burst identification and quantification.** Burst dynamics were analyzed using custom scripts following established methods in prior studies [54,77,98,122,123]. Bursts were identified by setting an ISI threshold of 12 ms, a commonly used value to distinguish intra-burst spikes from isolated spikes in ELL pyramidal cells [76,77]. Burst fraction was calculated as the ratio of spikes within bursts to the total spike count. ISI distributions were computed to quantify spiking variability and to characterize the firing patterns as bimodal, corresponding to bursts (short ISIs) and isolated spikes (long ISIs).

**Model optimization and parameter fitting.** Parameter fitting of the computational models to experimental data was performed in Python using the Optuna optimization framework [124]. We quantified the goodness of the fit using a weighted sum of normalized root mean squared errors (RMSE) across extracted features. This weighted error was used as the multi-objective loss function. Acceptable fits were defined as parameter sets with a total loss <0.1, corresponding to less than 10% normalized error across all features. For the biophysical model, optimization targeted both (i) action potential waveform features extracted from spike shape analyses (threshold, upstroke, peak, downstroke, trough, ADP) and (ii) spike-train metrics including coefficient of variation, ISI mean and median, mean firing rate, burst fraction, and standard deviation of ISIs. Model parameters were initially constrained within $\pm 20\%$ deviation ranges from previously published parameter values [25,39,56–58]. Optimization was carried out using the non-dominated sorting genetic algorithm II (NSGA-II) [125] implemented in Optuna.

For the modified HR model, we optimized parameters separately for each recorded cell using only spike-train features (mean firing rate, burst fraction, ISI mean, ISI median, ISI standard deviation, and ISI coefficient of variation). Waveform features were not included, consistent with the phenomenological nature of the HR formulation. Parameter sets were considered successful if the weighted loss was <0.1 and the simulated ISI distribution did not differ significantly from the experimental distribution. The successful parameter sets were used for the sensitivity analysis (Fig 7A). For comparisons of extracted features at the population level (Figs 3C, 6, and S3), model parameters were optimized separately for each recorded cell, resulting in distinct parameter sets that captured cell-specific spike features. For bifurcation analyses (Figs 4 and 5), the parameter set obtained from an example cell used to illustrate the dynamical structure is provided in

supporting materials (S1 and S2 Tables). For stochastic simulations comparing noisy and noise-free conditions (Fig 7B), parameters were fixed to those from the same example cell.

**Bifurcation and sensitivity analysis.** We performed bifurcation analyses using AUTO [126] to investigate dynamical transitions in the computational models. For the ELL pyramidal cell model, we conducted one-parameter bifurcation analysis with respect to the applied current ($I_{app}$) and automatically detected saddle-node and Hopf bifurcations (SN1, SN2, HB). To obtain branches of periodic orbits, we switched from the Hopf point (HB) to the small-amplitude limit cycle and continued that branch with smaller step sizes—$DS \in [0.5^{-2}, 1.0^{-4}]$, $DSMIN \in [10^{-3}, 10^{-4}]$, and $DSMAX \in [0.01, 1.0]$— to detect saddle-nodes of periodic orbits ($SNP_1$, $SNP_2$) and period-doubling ($PD_1$). Next, starting from these bifurcation points, we performed two-parameter continuations of the corresponding loci, pairing $I_{app}$ with either the SK or NMDA maximal conductances ($g_{SK}$ and $g_{NMDA}$, respectively) to explore model behavior systematically. Since fast–slow bursting orbits are stiff and have long periods near homoclinic bifurcations, we used (i) adaptive mesh refinement ($IADS = 3$) with mesh sizes $NTST \in [200, 500]$ and (ii) large caps on the maximum number of continuation steps $NMX \in [10^4, 10^6]$. In regions near bursting oscillations, we avoided detecting spurious singularities by setting $SP = ['LP0', 'BP0']$ to disable the detection of folds (LP) and branch points (BP). The relative tolerances were EPSL = EPSU = $10^{-8}$; for detection of special solutions, we used $EPSS \in [10^{-6}, 10^{-8}]$. Stability of periodic-orbit branches (solid vs. dashed) was determined from Floquet multipliers returned by AUTO.

For the modified HR model, we first executed bifurcation analysis over $I_{app}$, starting from the equilibria and continuing by varying $I_{app}$ to detect saddle-node bifurcations (SN1, SN2) and Hopf bifurcations (HB1 - HB6). followed by global sensitivity analysis using Sobol indices [127,128] to quantify the influence of individual parameters on firing patterns. To do so, we classified simulations obtained during parameter optimization across all ELL pyramidal cells in our dataset ($n = 32$) as "successful" if the total loss (e.g. the error measure between model output and experimental data) is below a threshold of $10^{-1}$ ($L < 0.1$). Simulations exceeding this threshold ($L \geq 0.1$) were excluded to ensure the analysis focused on parameter sets that yielded biologically plausible dynamics. For the retained successful simulations, we computed the total variance ($var_{total}$) in the loss metric and calculated the Sobol indices ($S_i$) using the equation:

$$S_i = \frac{var_i}{var_{total}} \tag{35}$$

where $var_i$ represents the variance attributable to parameter $i$. This approach allowed us to quantify the relative contribution of each parameter to the observed firing patterns.

**Software and computational resources.** All data analyses, simulations, feature extraction procedures, and parameter optimizations were performed using custom-written Python scripts executed on Compute Canada Alliance clusters. Bifurcation analyses were performed in AUTO-07 [129], with a custom script for visualization of results. Stochastic model simulations were integrated using the Euler–Maruyama method [130]. The excitatory and inhibitory pre-synaptic inputs in the biophysical ELL pyramidal cell model were generated using the algorithm described previously [119]. For the biophysical model, all simulations were run using an integration timestep of 0.1 ms (100 kHz sampling rate). The resulting membrane voltage traces were subsequently downsampled to 10 kHz to facilitate comparison with experimental recordings. This fine integration timestep was essential to ensure sufficient data points for accurate action potential shape analysis. For the modified HR model, simulations were performed at a 20 kHz sampling rate, after which the data were downsampled to 10 kHz to maintain consistency with experimental recordings.

## Code availability

Code used to run simulations, analyze data, and generate manuscript figures is available on GitHub (https://github.com/aminakhshi/spc_mobjective).

## Acknowledgments

The authors acknowledge the use of high-performance computing (HPC) resources provided by Calcul Québec (https://www.calculquebec.ca/) and the Digital Research Alliance of Canada (https://alliancecan.ca/) for this research.

## Supporting information

**S1 Fig. Quantification of *in vivo* burst firing statistics in ELL pyramidal cells. (A)** Illustration of burst definition and spike train separation for a representative neuron. Left: The burst threshold (dashed line) is set at the local minimum, separating short ISI mode (red) from longer ISIs associated with isolated spikes (green). Right: Spike raster plot of the same cell showing all recorded spikes (dark blue), and the resulting separation into burst spikes (red) and isolated spikes (green) based on the ISI threshold. **(B)** Distribution of mean firing rates (left, computed from all spikes over the entire recording duration) and burst fractions (right) for all recorded pyramidal cells ($N = 8$ fish, $n = 32$ cells). Each intracellular recording lasted 5 sec.
(TIFF)

**S2 Fig. Somatic and dendritic voltages and intracellular $Ca^{2+}$ dynamics. (A, B)** Somatic (A) and dendritic (B) membrane voltages. The somatic action potential backpropagates to the dendritic compartment, generating dendritic action potentials upon activation of $Na^+$ channels in the dendrite, which in turn propagates to the soma through electrodiffusion coupling. Spike train traces show a gradual decrease in consecutive ISIs during the burst period but no evidence of the depolarizing afterpotential (DAP) growth or failure in the dendritic action potentials upon the termination of the burst, similar to *in vivo* recordings. **(C, D)** Time series of the $Ca^{2+}$ concentration within the cytosol (C) and ER (D) from the dendritic compartment. $Ca^{2+}$ release into the dendritic membrane through NMDAR or via the ER activates SK channels, which in turn promotes afterhyperpolarization in the membrane potential. This mechanism is the primary driver of DAP termination, preventing prolonged dendritic spike backpropagation and accounting for the difference in the burst mechanism between *in vitro* and *in vivo* recordings.
(TIFF)

**S3 Fig. Comparison of action potential features between experimental data and biophysical model simulations across the population.** Scatter plots comparing the mean value of each action potential feature between experimental data (data) and corresponding fitted model simulations (model) for all recorded ELL pyramidal cells ($n = 32$). Features shown from left to right : spike threshold ($threshold_v$), midpoint of the upstroke phase ($upstroke_v$), peak amplitudes ($peak_v$), midpoint of the downstroke phase ($downstroke_v$), trough amplitudes ($trough_v$) and amplitude of afterdepolarization potential ($adp_v$). Points cluster closely around the identity line, indicating strong agreement (linear fit: $r_{threshold} = 0.98$, $p_{threshold} = 3.7 \times 10^{-23}$; $r_{upstroke} = 0.97$, $p_{upstroke} = 4.6 \times 10^{-21}$; $r_{peak} = 0.99$, $p_{peak} = 1.0 \times 10^{-27}$; $r_{downstroke} = 0.98$, $p_{downstroke} = 1.0 \times 10^{-26}$; $r_{trough} = 0.98$, $p_{trough} = 2.7 \times 10^{-24}$; $r_{adp} = 0.97$, $p_{adp} = 1.7 \times 10^{-20}$). Error bars indicate standard errors (SEM).
(TIFF)

**S4 Fig. Membrane potential simulations, and firing rate dynamics demonstrate transient action potential and different firing regimes in the biophysical model. (A)** Membrane potential simulations of the biophysical model (0.1 s duration) with identical parameter values but different initial conditions, using a subthreshold value of the depolarizing current ($I_{app}$) below the SN1 bifurcation point (see Fig 4A). Depending on the initial condition of the system, the model either returns directly to the resting state (right panel) or generates a single transient spike before returning to rest (left panel). **(B)** The firing rate of the biophysical two-compartment model detected within a time range of 20 seconds with respect to $I_{app}$, showcasing the chaotic dynamics exhibited by this model. The red curve represents the average firing rate. **(C)** Simulation of the biophysical model after blocking SK and NMDA currents $g_{SK} = 0$, $g_{NMDA} = 0$). In the absence of

these currents, the model exhibits prolonged bursts with prominent depolarizing afterpotentials (DAPs), reverting to dynamics characteristic of the original in vitro ghostbursting model.
(TIFF)

**S5 Fig. Initial condition dependence of subthreshold responses in the modified HR model.** Membrane potential simulations (1s duration) for the same parameter values using two different initial conditions with the applied current $I_{app}$ below the spiking threshold. Depending on the specific initial condition, the model either generates a single transient spike before returning to rest (left) or returns directly to the resting state without spiking (right).
(TIFF)

**S6 Fig. Comparison of the interspike interval distributions between an example recorded ELL pyramidal cell and model simulations of modified HR model.** Comparison of the interspike interval (ISI) distributions between an example recorded ELL pyramidal cell (blue) and ISIs from model simulations of modified HR model (teal, left) and classic HR model (purple, right) both having same parameter values and stochastic synaptic input (Kolmogorov-Smirnov test vs data: $D_{\text{modified HR}} = 0.146$, $p_{\text{modified HR}} = 0.778$; $D_{\text{HR}} = 0.383$, $p_{\text{HR}} = 0.0065$).
(TIFF)

**S1 Table. Parameter values used in the two-computational model.** Values with a range are determined individually for each cell by fitting.
(XLSX)

**S2 Table. Parameter values utilized in the deterministic, modified HR model for bifurcation analysis.**
(XLSX)

## Author contributions

**Conceptualization:** Amin Akhshi, Maurice J. Chacron, Anmar Khadra.

**Data curation:** Amin Akhshi, Michael G. Metzen.

**Formal analysis:** Amin Akhshi, Michael G. Metzen.

**Funding acquisition:** Maurice J. Chacron, Anmar Khadra.

**Methodology:** Amin Akhshi.

**Software:** Amin Akhshi.

**Supervision:** Maurice J. Chacron, Anmar Khadra.

**Writing – original draft:** Amin Akhshi, Anmar Khadra.

**Writing – review & editing:** Amin Akhshi, Michael G. Metzen, Maurice J. Chacron, Anmar Khadra.

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
