## [Decision Letter · Decision Letter 0]

14 Jul 2025

PCOMPBIOL-D-25-01154

In vivo neural activity of electrosensory pyramidal cells: Biophysical characterization and phenomenological modeling

PLOS Computational Biology

Dear Dr. Khadra,

Thank you for submitting your manuscript to PLOS Computational Biology. After careful consideration, we feel that it has merit but does not fully meet PLOS Computational Biology's publication criteria as it currently stands. Therefore, we invite you to submit a revised version of the manuscript that addresses the points raised during the review process.

Please submit your revised manuscript within 60 days Sep 13 2025 11:59PM. If you will need more time than this to complete your revisions, please reply to this message or contact the journal office at ploscompbiol@plos.org. Please include the following items when submitting your revised manuscript:

We look forward to receiving your revised manuscript.

Kind regards,

Renaud Blaise Jolivet, Ph.D.

Academic Editor

PLOS Computational Biology

Lyle Graham

Section Editor

PLOS Computational Biology

**Journal Requirements:**

At this stage, the following Authors/Authors require contributions: Amin Akhshi, Michael G. Metzen, Maurice J. Chacron, and Anmar Khadra. Please ensure that the full contributions of each author are acknowledged in the "Add/Edit/Remove Authors" section of our submission form.

3) Your manuscript is missing the following section: Abstract.  Please ensure all required sections are present and in the correct order. Make sure section heading levels are clearly indicated in the manuscript text, and limit sub-sections to 3 heading levels. An outline of the required sections can be consulted in our submission guidelines here:

5) We have noticed that you have uploaded Supporting Information files, but you have not included a list of legends. Please add a full list of legends for your Supporting Information files after the references list.

Potential Copyright Issues:

i) Figures 1A, and 2A. Please confirm whether you drew the images / clip-art within the figure panels by hand. If you did not draw the images, please provide (a) a link to the source of the images or icons and their license / terms of use; or (b) written permission from the copyright holder to publish the images or icons under our CC BY 4.0 license. Alternatively, you may replace the images with open source alternatives. See these open source resources you may use to replace images / clip-art:

7) Please amend your detailed Financial Disclosure statement. This is published with the article. It must therefore be completed in full sentences and contain the exact wording you wish to be published.

2) If any authors received a salary from any of your funders, please state which authors and which funders.

8) Please ensure that the funders and grant numbers match between the Financial Disclosure field and the Funding Information tab in your submission form. Note that the funders must be provided in the same order in both places as well. Currently, the order of this grant "Canada First Research Excellence Fund (HBHL scholarship" is different in both places.

**Reviewers' comments:**

Reviewer's Responses to Questions

**Comments to the Authors:**

**Please note that one of the reviews is uploaded as an attachment.**

Reviewer #1: See attachment

Reviewer #2: This manuscript combines intracellular recordings and computational modeling to investigate burst firing in ELL cells of weakly electric fish. A biophysically detailed model and a modified HR model reveal that in vivo burst firing arises from interactions among NMDA receptors, SK channels, calcium dynamics, and synaptic noise. The findings explain variability in firing patterns and help reconcile differences between in vivo and in vitro observations.

(1) With increasing Iapp, the biophysically detailed ELL cell model shows a gradual transition from quiescence to tonic firing and then to bursting (supplementary Fig 4B), whereas the modified HR model exhibits an abrupt change from quiescence to burst firing (fig 5B). I am curious what type of firing pattern do actual ELL cells exhibit with increasing Iapp?

(2) Related to the above: as stated in the manuscript, factors such as synaptic inputs and neuromodulators can modulate neuronal responses such as firing patterns. Is it possible that these factors could shift a single cell’s firing pattern between two firing patterns represented by biophysically detailed ELL cell model and modified HR model? Or shifting the intrinsic intrinsic neuronal mechanisms

(3) I understand that various types of models can be fitted to reproduce key features of a particular neuron type, and bifurcation analysis is often used to uncover potential intrinsic mechanisms of this type of neuron. The authors have explained why two models with distinct firing patterns are used to represent ELL cells in line 225-238. However, I’m still trying to understand why (and how) two different underlying mechanisms, demonstrated by the respective bifurcation diagrams of two models, contributes to understand in vivo behaviors of one type of neuron (i.e., ELL cells). A brief and straightforward explanation would help readers like me better grasp the motivation behind this approach. Additionally, a short summary comparing the common observations, differing conclusions, and limitations of the two models would be very helpful.

(4) Lines 267-269 states that the modified HR model shows sensitivity to initial condition. Additionally, the bifurcation analysis for both models has metastability regions. If I understand correctly, this suggests that the coexistence of multiple stable states depending on initial conditions. Does this sensitivity/dependence of initial conditions have any biophysical relevance to actual ELL cells or other cells in other systems in general?

(5) Line 321-322: “we fixed all parameters of the deterministic modified HR model based on the SAME EXAMPLE ELL pyramidal cell used earlier and simulated its activity with and without stochastic synaptic noise (Fig. 7B).” (a) Does this mean the authors selected a specific set of parameter values that best fit the recording of this particular example ELL cell? (b) For the biologically detailed ELL model in Figs 2 and 3, were a fixed set of parameter values used, or did the parameter values vary across the different recorded ELL cells (n=20)? (c) For modified HR model in Fig 6, were a fixed set of parameter values used, or did the parameter values vary across different recorded ELL cells? (d) If different sets of parameter values are used for different ELL cells, could the author clarify what the parameter values are used for bifurcation analysis in Figs 4 and 5?

(6) How is the amplitude of afterdepolarization potential (adp_v) in Fig 2D calculated for the first action potential (or the first several APs) in a burst (such as those in Fig. D3 and D4 for simulation) where the adp_v is not clearly observable in either simulation or recordings?

(7) Modified HR model on lines 640-643: (a) according to lines 655-656, dW terms in four equations (29)-(33) appear to be different, please correct the mathematical expressions to reflect this. (b) equation number (32) is missed.

Reviewer #3: This is a technically sound manuscript that addresses how to reconcile in vivo recordings from ELL pyramidal cells in weakly electric fish with in vitro data, using both biophysically detailed and phenomenological neural models. The electrosensory system is relatively small and well characterised, making it an excellent candidate for mechanistic modelling. The authors combine electrophysiological recordings, computational simulations, and bifurcation analyses to investigate how background synaptic activity, intracellular calcium dynamics, and membrane conductances interact to shape burst firing.

While the experimental, modelling, and bifurcation analyses appear to be carefully executed -- despite some issues noted below -- the manuscript still contains minor omissions, technical inconsistencies, and linguistic errors that require attention. A more thorough revision is needed to improve clarity and presentation. Overall, the simulations offer interesting insights into discrepancies between in vitro and in vivo ELL electrophysiology, particularly regarding bursting regimes. The proposed mechanism, based on the interplay between noise and multistable dynamics, provides an intuitive and potentially valuable framework for understanding this phenomenon. If the text is revised to address the specific points raised below, I believe it would meet the standards for publication in PLoS CB.

# Major

L13-15: Background synaptic activity does not necessarily lead to a depolarised bias. While it introduces membrane potential fluctuations and can enhance responsiveness, the net effect on membrane potential depends on the balance of excitation and inhibition. Typically, it leads to a high-conductance state with increased variability, not necessarily a shift in mean voltage. See Destexhe et al., 2003; Shu et al., 2003; Haider & McCormick, 2009; Richardson, 2004.

L98-104 and discussion on chaotic bursting. How do your results compare to [Falcke, M., Huerta, R., Rabinovich, M. I., Abarbanel, H. D., Elson, R. C., & Selverston, A. I. (2000). Modeling observed chaotic oscillations in bursting neurons: the role of calcium dynamics and IP 3. Biological cybernetics, 82, 517-527]?

Please provide more details on the numerical continuation: it is notoriously hard to continue fast-slow dynamics, such as bursting orbits, and yet you seem to track them just fine (dashed green lines on Figs 4,5).

L125, Fig. 2D: t-tests assess mean differences only, but the boxplots suggest full distributions are of interest. Consider also reporting distributional metrics or using nonparametric tests.

L189: please expand the discussion on ghostbursting and doublet firing: it is unclear how each are defined and arise from periodic orbit bifurcations: Fig 4D-3 even says “ghostbursting in the form of doublets”.

L555: Why is the NMDA driving force written as (V-VCa)? Standard NMDA reversal is ~0 mV. If this term refers to Ca++ influx through NMDA channels, please clarify that explicitly.

Fig. 5A: The dashed blue lines are not explained in the caption -- presumably unstable periodic orbits? It is also unclear what happens to the orbits emerging from HB6 as Iext increases; the branch appears to vanish without a visible bifurcation, despite persisting beyond HM2. Similarly, the period-2 orbits from PD1 and PD2 seem to disappear without a clear mechanism. The HR bifurcation diagram would benefit from clearer labelling of key regions and bifurcations, and improved visual design -- for instance, use shading instead of dashed boxes for insets, and revise colour choices for better contrast.

Fig. 5B: It is difficult to follow the f-I curve. Starting from I = 0 (for which the system seems monostable, even though it is hard to assess that given the previous comment) and tracking the quiescent branch (zero frequency) for increasing I, from the bifurcation diagram oscillations should emerge just before I = 0.5 (at HB1), but the f-I curve remains flat beyond I=0. This conflicts with Fig. 5C, which shows tonic firing at I = 0.4. If this discrepancy is due to multistability, it should be explicitly mentioned and clearly supported by the bifurcation diagram. Please revise the figure and caption to ensure consistency and clarify the dynamic regimes being shown.

L247-250: Please clarify what kind of multistability is being referred to. The bottom inset in Fig. 5A suggests that the stable periodic orbit (solid green) emerges from the stable equilibrium at HB3, so it cannot coexist with it -- i.e. not multistability between silence and tonic firing. Is the multistability between silence/tonic and the period-2 orbits from PD2 (in adjacent I ranges)? Or with the large-amplitude (light green) orbits emerging from the torus bifurcation? This region is hard to interpret (see earlier comments on Fig. 5A). Please revise this section carefully for clarity. The terminology and schematic representations of multistable regimes in [Marin, B., Barnett, W. H., Doloc-Mihu, A., Calabrese, R. L., & Cymbalyuk, G. S. (2013). High prevalence of multistability of rest states and bursting in a database of a model neuron. PLoS Computational Biology, 9(3), e1002930] could be of use. Consider adding a schematic summary of the dynamical regimes (e.g. quiescent, tonic, bursting) across parameter space to aid interpretation of the bifurcation diagrams. Consider adding a schematic summary of the dynamical regimes (e.g. quiescent, tonic, bursting) across parameter space to aid interpretation of the bifurcation diagrams.

Fig. 6, L301: “reproduces realistic spiking behavior” is vague. Were parameters in the HR model optimised to fit the data? What would count as a failure or unrealistic firing? Please clarify your criteria.

L483, elsewhere: This study does not rely on “analytical tractability,” nor does it include a systematic evaluation of model efficiency. Please avoid such statements unless they are supported by actual analysis or data.

The manuscript models background input as additive current noise, but under in vivo-like synaptic bombardment, noise is typically conductance-based and thus multiplicative in nature (e.g. Destexhe et al., 2003; Richardson, 2004). Please clarify whether the model approximates this, and whether the implications of state-dependent noise were considered.

# Minor

## Abstract:

"calcium handling" sounds vague.

## Summary:

"This simplified phenomenological model successfully captured burst firings" sounds awkward: use either "bursting" or "burst firing" (singular).

“validate simplified frameworks for studying population-level dynamics” the link between what has been discussed in the paper and population dynamics is not entirely clear.

## Main text

L3: “given neural class” please clarify.

L35: please clarify “dendritic failure”.

L49: “constrained its parameter regimes” sounds off - maybe parameter sets?

L52: Hindmarsh-Rose is a model, not a formalism - suggest avoiding that term.

L53: “significant advantages for modeling population coding…” feels overstated. The HR model is certainly efficient as a point neuron with mostly polynomial dynamics, but the actual impact on large-scale simulations is not demonstrated here, especially given the sigmoid dynamics in the fourth variable.

L57, 58: just one of the black arrows corresponds to EA->EEL.

L63: which kind of stimulation? current clamps?

L63: “(Fig. 1A, left; see Methods for details)” shouldn't it be 1A, *right*?

L111: “identify the physiologically relevant parameter ranges in vivo” is a bit clunky - suggest rephrasing for clarity.

L168 “how their effect on” ?

L170: please define "ghostbursting".

L182: “two envelopes of stable limit cycle” is clunky and conceptually off - the stable limit cycle branch emerges from the Hopf bifurcation; the upper/lower bounds are just amplitude envelopes shown in the bifurcation plot.

L185: similar as above: it is not “the envelopes” that undergo period doubling, but the periodic orbit.

L201 “boundaries of *the* bursting region”: add article

L224, 230, 350, 632: Again, I do not think Hindmarsh Rose is a “formalism”: it is just a model.

L236: “gamma-frequency burst oscillations” have not been defined or mentioned before: are those observed in vivo?

L244 “a branch of _an_ unstable equilibria”: remove “an”

L246: see discussion on the use of “envelopes” for L182; please revise elsewhere as well.

L280 “we have added extracellular noise term”: missing article

L332 “Specifically, it exhibits a rightward shift” do you mean skewness?

L358:”dendrites This”: missing period.

L429: could multistability also lead to dysfunctional regimes? (see eg [Marin, B., Barnett, W. H., Doloc-Mihu, A., Calabrese, R. L., & Cymbalyuk, G. S. (2013). High prevalence of multistability of rest states and bursting in a database of a model neuron. PLoS Computational Biology, 9(3), e1002930]

L576: Does Eq. [14] truly correspond to an alpha function? It looks like a single exponential decay, without the rising phase typical of alpha functions.

L611 "[...] the power spectra [...] exhibit a power law spectrum": rephrase

Eqs. 29-33 should use SDE notation: dx = f(x)dt + σdW. Writing dx/dt = f(x) + dW is not mathematically valid, since dW/dt is undefined. Also, the dWi are different for each variable. See next comment as well.

Line 655: dWi is not a Wiener process, but the differential of one, i.e. stochastic increments. Suggest clarifying that Wi(t) are Wiener processes and dWi are their increments used in the SDEs.

L725: "Model simulations were carried out using the Euler-Maruyama method (Kloeden et al., 1992)as the integration solver of the stochastic models.": awkward phrasing.

Fig. 1A: "higher brain area" sounds vague. Consider "higher-order brain region" or a specific anatomical label.

Fig. 1B: The phase plane plot seems of limited value - consider replacing it with a zoomed-in view of superimposed spikes, which would better illustrate spike shape variability. Also, the Vm trace appears low-resolution (at least in the PDF), making it hard to assess waveform features like ADPs and AHPs.

L757 "for a representative electrophysiological data from an ELL pyramidal cell.": "a representative electrophysiological data" is ungrammatical. Suggest "representative data" or "a representative recording".

Fig 2C: maybe zoom around spike so that features in B are be more visible? try playing with transparency / line widths so that experimental traces are more discernible?

L765: Gray box? Markov model? Please revise figure / legend.

L773: (rough_v) -> (Trough_v)

L777: “modeling simulation” -> model simulation / simulation of the model

L778: Stray comma/parenthesis after “threshold_v”. Also, “rough_v” -> trough_v

Fig 3A: same panel as in fig 1B; all Vm traces look low-res and hard to interpret - the only thing I can almost clearly extract from these plots is the same information provided by the rasters in B (spike times).

Fig 3C, 6C: consider using unfilled circles (or smaller markers?); also, pay attention to label / scale fontsizes throughout the figure

Fig 4A, right: aren’t PD and PD1 the same bifurcation? also, where is PD2 in the top branch? it is hard to follow periodic orbit branch at this zoom level, maybe it would be helpful to show a zoomed in version of just one part (max or min) of it?

L800: the model undergoes bifurcations (remove “points”)

L802: “regimes of behavior”?

L803: There is only one “bounding box”; consider rephrasing as “dashed box” or “highlighted region”.

L804/807: “the two parameter bifurcation ___?” missing “diagram”?

L805/808: “highlighting the various regions of behavior that the model possesses” clunky.

L806: “Inherently multistable” is vague. Multistability arises from coexisting attractors, not as an inherent property of a region bounded by a bifurcation set. The rest of the sentence (“it overlaps with other regions”) is also confusing. Please clarify what regions are meant and how this relates to multistability.

L811: different values of Iapp, gSK and gNMDA, right? maybe refer to the points 1,2,3,4 highlighted in parameter space?

Fig 5: Is any part of the model actually dimensional? You use a.u. for the V variable and for I, but seconds and Hz for time and frequency.

Fig 6 and others: would decreasing line widths for all time series lead to clearer plots?

**Have the authors made all data and (if applicable) computational code underlying the findings in their manuscript fully available?**

Reviewer #1: Yes

Reviewer #2: Yes

Reviewer #3: Yes

PLOS authors have the option to publish the peer review history of their article (what does this mean?). If published, this will include your full peer review and any attached files.

Reviewer #2: No

Reviewer #3: No

**Figure resubmission:**
---

## [Decision Letter · Decision Letter 1]

6 Nov 2025

Dear Dr. Khadra,

We are pleased to inform you that your manuscript 'In vivo neural activity of electrosensory pyramidal cells: Biophysical characterization and phenomenological modeling' has been provisionally accepted for publication in PLOS Computational Biology.

Best regards,

Renaud Blaise Jolivet, Ph.D.

Academic Editor

PLOS Computational Biology

Lyle Graham

Section Editor

PLOS Computational Biology

Please make sure to address the final few comments from Reviewer 3 prior to submitting the final files.

Reviewer's Responses to Questions

**Comments to the Authors:**

Reviewer #1: I believe the manuscript in its new form has been substantially improved by the authors in response to its reviewers, in particularly in regards to its initial lack of clarity on some of the experimental points.

I recommend the manuscript for publication and thank the authors for their contribution.

Reviewer #2: The authors have satisfactorily addressed all my previous comments. The revisions have significantly improved the clarity and quality of the manuscript. I have no further concerns and recommend acceptance in its current form.

Reviewer #3: The authors have satisfactorily addressed my previous comments.

I only have a few remaining minor suggestions:

# Figure 4 caption

"1 Hopf bifurcation_s_ (HB)": singular.

"2 saddle-node of _periodics_": replace with 'periodic orbits'.

"multistable with distinct attractors coexisting under the same parameter values, depending on initial conditions": the system is multistable for these parameter values, regardless of initial conditions. You will end up seeing one of the attractors depending on the IVP solved. Rephrase.

# Figure 5 caption

"2 saddle-node of _periodics_": replace 'periodic orbits'.

"2 homoclinic bifurcations (HM1, HM2)": a third one (HM3) appears in the bottom inset.

"detected within a time range of 20 seconds long": rephrase.

"showcasing the chaotic dynamics exhibited by this model similar to the biophysical model": rephrase.

# Main text

L373: "ghostbu_sting": the pun is nice but probably unintended

L937 "Floccules"?

**Have the authors made all data and (if applicable) computational code underlying the findings in their manuscript fully available?**

Reviewer #1: Yes

Reviewer #2: None

Reviewer #3: Yes

PLOS authors have the option to publish the peer review history of their article (what does this mean?). If published, this will include your full peer review and any attached files.

Reviewer #1: **Yes: **Niccolo' Calcini

Reviewer #2: No

Reviewer #3: No

---

## [Editor Report · Acceptance letter]

PCOMPBIOL-D-25-01154R1

In vivo neural activity of electrosensory pyramidal cells: Biophysical characterization and phenomenological modeling

Dear Dr Khadra,

I am pleased to inform you that your manuscript has been formally accepted for publication in PLOS Computational Biology. Your manuscript is now with our production department and you will be notified of the publication date in due course.

With kind regards,

Anita Estes
